# EAGLE: EXPLORING THE DESIGN SPACE FOR MULTIMODAL LLMS WITH MIXTURE OF ENCODERS

**Min Shi**[2*]**, Fuxiao Liu**[3*]**, Shihao Wang**[4]**, Shijia Liao**[1]**, Subhashree Radhakrishnan**[1]**,
Yilin Zhao**[5]**, De-An Huang**[1]**, Hongxu Yin**[1]**, Karan Sapra**[1]**, Yaser Yacoob**[3]**, Humphrey Shi**[2]**,
Bryan Catanzaro**[1]**, Andrew Tao**[1]**, Jan Kautz**[1]**, Zhiding Yu**[1†]**, Guilin Liu**[1†]

[1]NVIDIA    [2]Georgia Tech    [3]UMD    [4]HKPU    [5]NYU
https://github.com/NVlabs/Eagle

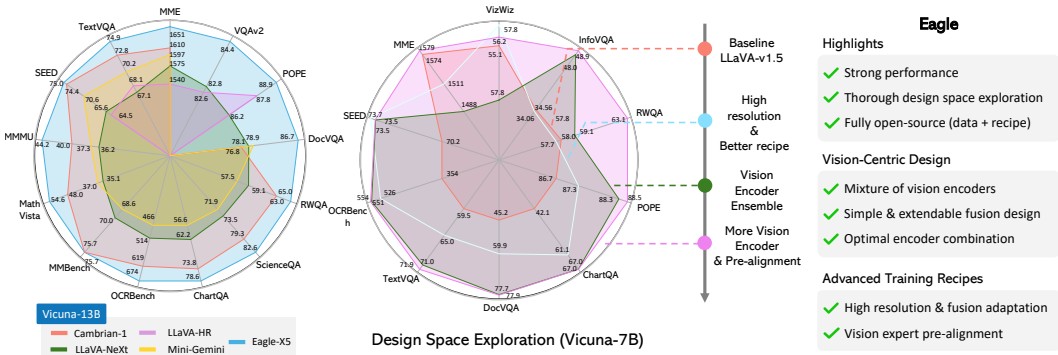

Figure 1: **Overview of *Eagle*.** *Eagle* is a family of multimodal large language models (MLLMs) with a mixture of vision encoders. Left: comparisons between *Eagle* and existing competitive MLLMs with *Vicuna-13B* (Chiang et al., 2023), with *Eagle* achieving favorable results on all 13 benchmarks. Middle: an evolutionary road map of the design space and advanced training recipes leading to consistent and significant improvements. Right: highlights and core features of *Eagle*.

## ABSTRACT

The ability to accurately interpret complex visual information is a crucial topic of multimodal large language models (MLLMs). Recent work indicates that enhanced visual perception significantly reduces hallucinations and improves performance on resolution-sensitive tasks, such as optical character recognition and document analysis. A number of recent MLLMs achieve this goal using a mixture of vision encoders. Despite their success, there is a lack of systematic comparisons and detailed ablation studies addressing critical aspects, such as expert selection and the integration of multiple vision experts. This study provides an extensive exploration of the design space for MLLMs using a mixture of vision encoders and resolutions. Our findings reveal several underlying principles common to various existing strategies, leading to a streamlined yet effective design approach. We discover that simply concatenating visual tokens from a set of complementary vision encoders is as effective as more complex mixing architectures or strategies. We additionally introduce *Pre-Alignment* to bridge the gap between vision-focused encoders and language tokens, enhancing model coherence. The resulting family of MLLMs, *Eagle*, surpasses other leading open-source models on major MLLM benchmarks.

## 1 INTRODUCTION

The success of large language models (LLMs) has triggered significant interest in enabling their visual perception capability, such that they could see, understand, and reason in the real world. At the center of these multimodal large language models (MLLMs) (Fei et al., 2024) is a typical design

---

*Equal contribution. Work done during an internship at NVIDIA.
†Equal advising. Corresponding authors: {guilinl,zhidingy}@nvidia.com.

where images are converted into a series of visual tokens by the vision encoders and appended with the text embeddings. *CLIP* (Radford et al., 2021) is often chosen as the vision encoder since its visual representation is aligned with the text space by pre-training on image-text pairs. Depending on the architectures, training recipes, and the way how vision tokens are injected into the language model, there exist various notable families of MLLMs such as *Flamingo* (Alayrac et al., 2022), *BLIP* (Li et al., 2022; 2023d; Dai et al., 2024), *PaLI* (Chen et al., 2023e), *PaLM-E* (Driess et al., 2023) and *LLaVA* (Liu et al., 2023d;c). Most of these works keep relatively low input resolutions due to the limits on pre-trained vision encoders and LLM sequence length.

Recent studies (Li et al., 2024c; Liu et al., 2024a) show that stronger vision encoder design is important for mitigating MLLM hallucinations (Liu et al., 2023a; Wu et al., 2024) and improving resolution-sensitive tasks like optical character recognition (OCR). A constellation of works thus focuses on enhancing the capability of the vision encoder. For example, scaling up the pre-training data and parameters of vision encoder (Chen et al., 2023f) or dividing images into low-resolution patches (Liu et al., 2024a; Shi et al., 2024). However, these approaches usually introduce large training resources. An efficient yet powerful strategy is to mix visual encoders pre-trained with different tasks and input resolutions, either fusing higher resolution encoders with the *CLIP* encoder (Luo et al., 2024; Li et al., 2024b), sequentially appending features from different encoders (Fan et al., 2024; Lin et al., 2023b; Karamcheti et al., 2024; Tong et al., 2024), or adopting more complex fusion and routing strategies to make the best of different encoders (Lee et al., 2024; Zong et al., 2024). In addition, Prismatic VLM (Liu et al., 2024b) incorporates multiple vision encoders with channel concatenation as part of their design space exploration. These "mixture-of-vision-experts" strategies are shown to be effective. However, a detailed study focusing on their designs is still lacking.

To address the above questions, our work systematically investigates the mixture-of-vision-encoders design space for improved MLLM perception. As shown in Fig. 1, our exploration of the design space consists of the following steps: 1) Benchmarking various vision encoders and searching recipes for higher resolution adaptation; 2) "Apples to apples" comparison between vision encoder fusion strategies; 3) Progressive identification of the optimal combination of multiple vision encoders; 4) Improved vision expert pre-alignment and data mixture. Our study covers the performance of vision encoders pre-trained on different tasks and resolutions (e.g., vision-language alignment (Ilharco et al., 2021; Cherti et al., 2023; Radford et al., 2021; Schuhmann et al., 2022), self-supervised learning (Oquab et al., 2023), detection (Fang et al., 2023b;a), segmentation (Kirillov et al., 2023), and OCR (Lee et al., 2023)). We use a round-robin approach to incorporate additional vision experts. Starting with the basic *CLIP* (Radford et al., 2021) encoder, we add one additional expert each time with the best improvement in each round.

Our work is not the first one to leverage multiple vision encoders in MLLM. However, the systematic study leads to several interesting new findings under this setting:

- Unlocking the vision encoders during MLLM training matters. This is in sharp contrast to the LLaVA (Liu et al., 2023d;c) family and many works that consider multiple vision encoders or teachers (Lin et al., 2023b; Liu et al., 2024b; Fan et al., 2024; Kar et al., 2024; Ranzinger et al., 2024; Lee et al., 2024), where freezing the vision encoders has been a common choice.

- Some recently proposed fusion strategies (Luo et al., 2024; Li et al., 2024b) do not show significant advantages despite their advanced designs. Instead, we find that straightforward channel concatenation stands out as a simple yet competitive fusion strategy, offering the best efficiency and performance.

- Incorporating additional vision experts leads to consistent gain, making it a promising path to systematically enhance MLLM perception besides scaling up vision encoders. The improvement is particularly pronounced when vision encoders are unlocked.

- We propose a pre-alignment stage where non-text-aligned vision experts are individually fine-tuned with a frozen LLM before being trained together. This stage is found to enhance the MLLM performance significantly under the mixture-of-vision-encoder design.

We finally conclude our findings into a family of MLLMs termed *Eagle*. *Eagle* is evaluated on a series of benchmarks, including visual question answering, OCR/document-related tasks, and benchmarks tailored for MLLMs. Our model attains state-of-the-art performance across different benchmarks and demonstrates obvious advantages on OCR and document understanding tasks. Using

the same pre-train and supervised fine-tuning data from *Cambrian-1* (Tong et al., 2024) - a concurrent family of vision-centric MLLMs sharing similar design spirits, *Eagle* models overall achieve better performance. We hope that the *Eagle* can provide a highly performant and easy-to-reproduce MLLM solution to the community.

## 2 DESIGN SPACE EXPLORATION

In this section, we show how to utilize the advantages of different vision encoders via step-by-step investigations, yielding the *Eagle* model family. Unlike previous methods focusing on new fusion strategies or architectures among vision encodes, our goal is to identify a set of minimalistic design to fuse different vision encoders supported with detailed ablations, removing any unnecessary parts. As shown in Fig. 2, we start by extending the basic *CLIP* encoder (Radford et al., 2021) to a set of vision experts with different architectures, pre-training tasks, and resolutions. With these experts, we then compare different fusion architectures and methods and study how to optimize the pre-training strategies given more encoders. We also give a detailed analysis of how to select the vision encoders to be integrated. Finally, we put all the findings together and further extend to multiple expert vision encoders with different resolutions and domain knowledge.

### 2.1 BASE SETUP

We adopt *LLaVA-1.5*'s (Liu et al., 2023c) model architecture as the basis, which consists of a large language model (Vicuna-v1.5 7B (Chiang et al., 2023)), a vision encoder, and a projection layer. The projection layer projects the visual embedding from the vision encoder into the text embedding space.

**Base training data.** We adopt the same pre-training data (**LLaVA-595k**) as *LLaVA-1.5* (Liu et al., 2023c) for the first pre-training stage, which consists of $595k$ image text pairs. To fully examine the potential of different vision experts and fusion methods, instead of using the SFT data from *LLaVA-1.5* (Liu et al., 2023c), we collect data from a series of tasks and convert them into multimodal conversations for the supervised fine-tuning (SFT) stage, denoted as **Eagle1.8M** in Table 1.

Table 1: Composition of the base supervised fine-tuning data (Eagle1.8M).

| Total Data Size | Data Source |
|---|---|
| 1,809k | *LLaVA*-1.5 (665k) (Liu et al., 2023c), DocVQA (39k) (Mathew et al., 2021), synDog-EN (50k) (Kim et al., 2022), ChartQA (28k) (Masry et al., 2022), DVQA (25k) (Kafle et al., 2018), AI2D (15k) (Kembhavi et al., 2016a), ShareGPT-4V (100k) (Chen et al., 2023b), laion-GPT4V (11k) (lai, 2023), LVIS-Instruct4V (220k) (Wang et al., 2023a), LRV-Instruct (150k) (Liu et al., 2023b), Geo170k (120k) (Gao et al., 2023), LLaVAR (20k) (Zhang et al., 2023), Visual7W (70k) (Zhu et al., 2016), Open-Hermes 2.5 (300k) (Teknium, 2023) |

**Base training recipes.** We start from the *LLaVA-1.5* (Liu et al., 2023c) recipe where the model is first pre-trained with image-text pairs for one epoch with a batch size of 256. The whole model is frozen and only the projector layer is updated in this pre-training stage. In the second stage, we further fine-tune the model on the multi-modal conversation data for one epoch with a batch size of 128. The learning rates are set to be *1e-3* for the first stage and *2e-5* for the second stage, respectively.

**Base evaluation.** To conduct a comprehensive comparison of various methods, we adopt 11 distinct benchmarks that span multiple tasks. These benchmarks include 1) **General VQA tasks**: GQA (Hudson & Manning, 2019), VizWiz (Gurari et al., 2018), MME (Fu et al., 2023), SEED (Li et al., 2023c); 2) **OCR/document/chart understanding**: OCRBench (Liu et al., 2023f), DocVQA (Mathew et al., 2021), ChartQA (Masry et al., 2022); 3) **vision-centric tasks**: POPE (Li et al., 2023e), Real-WorldQA (xAI, 2024); 4) **knowledge-based tasks**: ScienceQA (Saikh et al., 2022), AI2D (Kembhavi et al., 2016b). To obtain an average score, we normalize each benchmark to a total score of 1,000 and then calculate the average score across all benchmarks.

### 2.2 STRONGER *CLIP* ENCODER

We start our exploration by upgrading the vanilla *CLIP* Radford et al. (2021) model since it has become a standard choice for most of the MLLMs Liu et al. (2023d;c). While *CLIP* models are known to benefit multimodal tasks via the text-image alignment, they also have inherent drawbacks. For instance, many existing MLLMs (Liu et al., 2023c) tend to use the pre-trained *CLIP* resolutions (such as $224 \times 224$ or $336 \times 336$) as their input resolutions. In these cases, the encoders often fail to capture fine-grained details that are important for resolution-sensitive tasks like OCR and document understanding (Li et al., 2024c).

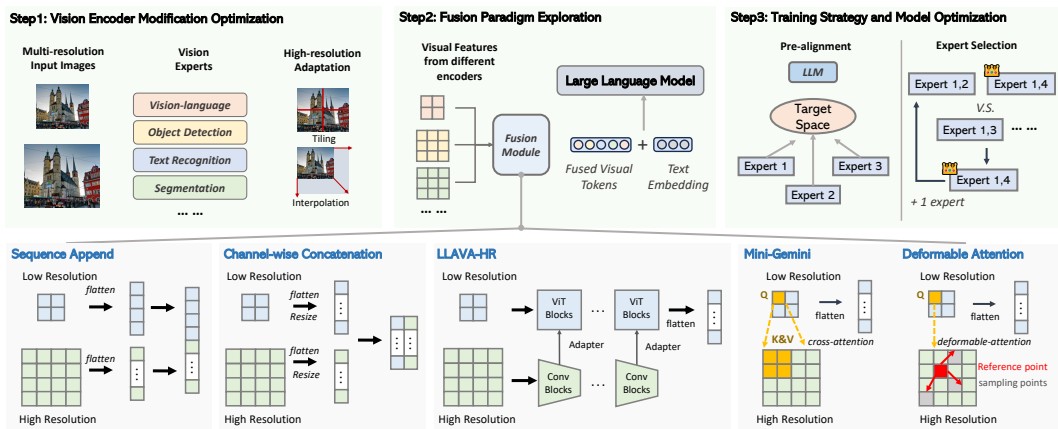

Figure 2: **Overview of the *Eagle* exploration**. In this work, we explore the design space of Multi-Modal Large Language Models (MLLMs) with multiple vision encoders, aiming to identify optimized design choices and enhance MLLM perception. We collect a range of vision experts and adapt them for integration into MLLMs. A systematic comparison of popular fusion paradigms is then conducted under controlled settings. After identifying discrepancies between vision experts pre-trained on different tasks, we optimize the pre-training strategy through a pre-alignment stage and use round-robin searching to determine the optimal combination of vision encoders.

To handle increased input resolution, a common practice is to use tiling where input images are divided into tiles and encoded separately (Liu et al., 2024a; Li et al., 2024c), or just interpolate the position embedding of the vision transformer model to fit high-resolution inputs (Chen et al., 2023c;d; Beyer et al., 2024). We compare these two approaches with frozen/unfrozen vision encoders under different resolutions, with the results shown in Table 2. Our findings can be summarized as follows:

- Updating the *CLIP* encoder during SFT significantly improves performance at higher resolutions but slightly reduces it when using the pre-training resolution.

- Interpolating CLIP encoder to fit the input size of $448 \times 448$ offers a strong balance between efficiency and performance, trailing the $672 \times 672$ version with less than half the tokens.

- Despite its smaller size (0.3B vs. 5.9B) and less pre-training data, the *CLIP* encoder gets close with interpolation approaches *InternVL*'s (Chen et al., 2023f) performance under the same setting.

Based on the results, we can see that *Direct interpolation* to $448 \times 448$ can achieve competitive performance while being more efficient. We thus use the *CLIP* encoder with a $448 \times 448$ input resolution while unlocking the encoder during the SFT stage.

Table 2: **Comparing different high-resolution adaption methods.** "#Tok(V)" denotes the number of visual tokens. "#Parames", "FLOPs" and "Img/Sec" denote the model size, complexity and throughput (bs=4) of the vision encoder.

| Method | Unfreeze | Res. | #Tok(V) | #Params | FLOPs | Img/Sec | Avg. |
|---|---|---|---|---|---|---|---|
| *Original* | ✗ | 336 | 576 | 0.3B | 119G | 197.2 | 616.5 |
| *Original* | ✓ | 336 | 576 | 0.3B | 119G | 197.2 | 562.6 |
| *Interpolate* | ✗ | 448 | 1024 | 0.3B | 214G | 119.5 | 589.7 |
| *Interpolate* | ✓ | 448 | 1024 | 0.3B | 214G | 119.5 | 670.5 |
| *Interpolate* | ✓ | 672 | 2304 | 0.3B | 480G | 56.3 | **674.2** |
| *Tiled-input* | ✓ | 672 | 2304 | 0.3B | 476G | 51.6 | 673.9 |
| *InternVL* | ✗ | 448 | 1024 | 5.9B | 5669G | 13.52 | 661.9 |
| *InternVL* | ✓ | 448 | 1024 | 5.9B | 5669G | 13.52 | 671.5 |

## 2.3 VISION EXPERTS

To better establish the foundation for multi-vision expert fusion, we extend the toolbox with vision experts pre-trained on different tasks and resolutions, and verify our findings on high-resolution adaptation with these experts. This also helps us identify the distinct advantages of different experts. We collect a set of vision experts, including: *(1) Vision-Language Alignment: CLIP* (Radford et al., 2021) and *ConvNeXt* (Liu et al., 2022) from *OpenCLIP* (Ilharco et al., 2021; Schuhmann et al., 2022). *(2) Object-Centric: EVA-02* (Fang et al., 2023b;a) pre-trained on detection datasets. *(3) OCR: Pix2Struct* (Lee et al., 2023).*(4) Segmentation: SAM* (Kirillov et al., 2023). *(5) Self-supervised: DINOv2* (Oquab et al., 2023). We resize the output 2D feature maps of each vision encoder using bilinear interpolation or pixel shuffle (Shi et al., 2016) to ensure that the visual token number equals 1024.

Results in Table 3 show that unfreezing the vision experts again leads to consistent improvement, which is aligned with Sec. 2.2. In addition, results in Table 10 (see Appendix A.1) further demonstrate that ***MLLMs with these task-specific vision encoders achieve optimal performance in their pre-training domains***. *EVA-02* excels in the object hallucination evaluation benchmark POPE and general visual question answering benchmark GQA. *CLIP* and *ConvNeXt*

Table 3: **Comparison between different vision experts as the MLLM encoders.**

| Category | Vision Encoder | Res. | Post-process | Unfreeze | Avg. | Model Link |
|---|---|---|---|---|---|---|
| *VL Alignment* | *ConvNeXt* | 1024 | None | ✗ | 654.6 | ConvNeXt-XXL |
| | | | | ✓ | **682.1** | |
| *Segmentation* | *SAM* | 1024 | Pixel Unshuffle | ✗ | 486.2 | SAM-ViT-Large |
| | | | | ✓ | **510.5** | |
| *Object Detection* | *EVA-02* | 1024 | Resize | ✗ | 543.7 | EVA-02-L-Det |
| | | | | ✓ | **639.1** | |
| *Text Recognition* | *Pix2Struct* | 1024 | Resize | ✗ | 598.6 | Pix2Struct-02-Large |
| | | | | ✓ | **606.2** | |
| *Self-Supervised* | *DINOv2* | 448 | None | ✗ | 520.7 | ViT-L/14-Reg |
| | | | | ✓ | **537.3** | |

perform well across all benchmarks, benefiting from their training on large-scale image-text pairs using contrastive loss. Conversely, while *Pix2Struct* excels in text recognition, it shows limited capability in object recognition and general VQA tasks, like POPE and GQA. *DINOv2* and *SAM*, pre-trained with self-supervised learning and segmentation, struggle with text recognition tasks.

## 2.4 FUSION STRATEGY

Existing MLLM frameworks have proposed various mixture-of-vision-encoder strategies, with the hope that their domain-specific strengths can be leveraged. In all cases, improvements in MLLM performance have been reported with the fusion of vision encoders. However, the roles of the fusion strategies as part of their MLLM architecture innovations, have not been decoupled and clearly studied under an "apples to apples" comparison. It is thus not entirely clear how much improvement is from the fusion strategies themselves versus the improved representations from various encoders.

We notice that existing popular fusion strategies, despite their variations in designs, can be broadly represented by the following several categories: (1) *Sequence Append*: directly appending the visual tokens from different backbones as a longer sequence (Fan et al., 2024; Kar et al., 2024); (2) *Channel Concatenation*: concatenating the visual tokens along the channel dimension without increasing the sequence length (Lin et al., 2023b; Karamcheti et al., 2024); (3) *LLaVA-HR*: injecting high-resolution features into low-resolution vision encoders using mixture-of-resolution adapter (Luo et al., 2024); (4) *Mini-Gemini*: using the *CLIP* tokens as the low-resolution queries to cross-attend another high-resolution vision encoder in the co-located local windows (Li et al., 2024b). (5) *Deformable Attention*: a new baseline we introduce on top of *Mini-Gemini*, where the vanilla window attention is replaced with deformable attention (Zhu et al., 2021). Fig. 2 gives a detailed illustration of these fusion strategies. To better study them, we choose "*CLIP+ConvNeXt*" and "*CLIP+ConvNeXt+SAM*" as the base multi-encoder combinations to perform comparisons.

Our study in Table 4 shows that *Channel Concatenation* stands out with the best performance, expandability, and efficiency. The "injection-based" methods, such as *LLaVA-HR*, *Mini-Gemini* and *Deformable Attention*, are in general less competitive on TextVQA (Singh et al., 2019) and OCR-Bench (Liu et al., 2023f), performing worse than using *ConvNeXt* alone as the vision encoder. Although sequence append shows comparable performance to channel concatenation, it faces the challenge of handling significantly increased sequence lengths with additional vision encoders.

Table 4: **Comparison of different fusion methods for different vision experts.** "#Token(V)" denotes the number of visual tokens. "#Tokens/s" denotes the inference throughput of the whole pipeline.

| Vision Encoders | Fusion | #Token(V) | #Tokens/s | #Params | Avg. |
|---|---|---|---|---|---|
| | *Seq. Append* | 2048 | 46.1 | 1200M | **690.5** |
| | *Channel Concat.* | 1024 | **47.3** | **1184M** | 681.5 |
| *CLIP + ConvNeXt* | *LLaVA-HR* | 1024 | 47.0 | 1219M | 678.7 |
| | *Mini-Gemini* | 1024 | 45.3 | 1201M | 672.5 |
| | *Deformable Attn.* | 1024 | **47.3** | 1201M | 674.3 |
| *CLIP + ConvNeXt* | *Seq. Append* | 3072 | 40.3 | 1529M | 686.2 |
| *+ SAM* | *Channel Concat.* | 1024 | **46.3** | **1495M** | **690.4** |

## 2.5 VISION-LANGUAGE PRE-ALIGNMENT

As shown in Table 3, encoders pre-trained exclusively on vision tasks (*e.g.*, detection, OCR, and segmentation) are less competitive compared to those pre-trained on vision language alignment. This is possibly due to representational inconsistencies when integrated with large language models.

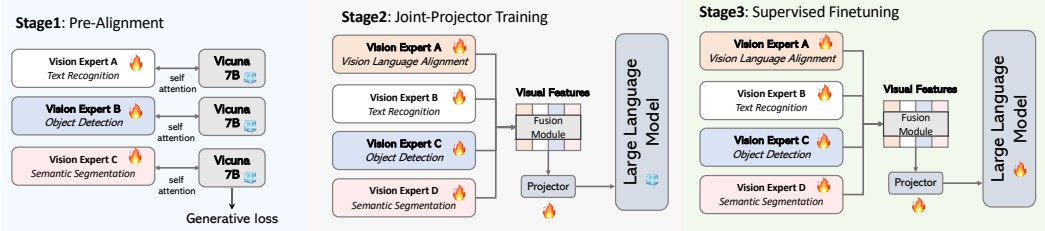

Figure 3: **The proposed training strategy of *Eagle*.** It consists of three progressive stages, including *vision-language pre-alignment training*, *joint-project training* and *supervised fine-tuning*. These stages effectively leverage public data from diverse sources, ranging from noisy image-text pairs on the web to high-quality caption, VQA, and multimodal dialogue datasets.

Additionally, when combining different encoders, there is a gap between these encoders, creating difficulties in the training process. To address this feature inconsistency, we propose a *Pre-Alignment* training stage that first aligns each individual vision encoder with the same large language model, fostering better synergy between visual and linguistic capabilities.

Fig. 3 depicts our pre-alignment strategy. Instead of training a projector to simultaneously align multiple vision experts as in *LLaVA*'s (Liu et al., 2023c) original pre-training strategy, we first align the representation of each individual expert with a smaller language model (Vicuna-7B in practice) using next-token-prediction supervision. As shown in Fig. 3, with pre-alignment, the whole training process consists of three steps: 1) *training each pre-trained vision expert with their own projector, while keeping the language model frozen;* 2) *combining all vision experts from the first step and training both the projector and vision experts;* 3) *training the whole model on SFT data.*

To verify the proposed method, we compare the pre-alignment strategy with the normal two-stage training strategy in Table 5, considering both freezing and unfreezing vision experts for comparison. As shown in Table 5, although unfreezing the vision experts during SFT helps improve performance by updating the vision experts to fit the language model, the *Pre-Align* strategy more effectively mitigates the inherent biases of each vision expert and stabilizes the training process, subsequently improving overall performance.

Table 5: **The effectiveness of *Pre-alignment*.**

| CLIP | Vision Expert (X) | Unfreeze | Pre-align | Avg. |
|---|---|---|---|---|
| *CLIP-448* | *SAM-1024* | ✗ | ✗ | 630.6 |
| | | ✗ | ✓ | 648.5 |
| | | ✓ | ✗ | 662.5 |
| | | ✓ | ✓ | **672.3** |
| *CLIP-448* | *ConvNext-1024* | ✗ | ✗ | 652.0 |
| | | ✗ | ✓ | 670.1 |
| | | ✓ | ✗ | 681.5 |
| | | ✓ | ✓ | **686.2** |
| *CLIP-448* | *Pix2Struct-1024* | ✗ | ✗ | 653.5 |
| | | ✗ | ✓ | 665.7 |
| | | ✓ | ✗ | 673.7 |
| | | ✓ | ✓ | **680.4** |
| *CLIP-448* | *EVA-02-L-1024* | ✗ | ✗ | 630.2 |
| | | ✗ | ✓ | 645.2 |
| | | ✓ | ✗ | 659.2 |
| | | ✓ | ✓ | **668.2** |

## 2.6 EXTENSION TO MULTI-EXPERTS

With the optimized strategies and training recipes of incorporating individual vision experts, we consider the incorporation of even more vision experts to push the limit. To conduct the search in a systematic manner, we adopt a step-by-step greedy strategy to incorporate additional vision experts.

We consider the vision experts discussed in Section 2.3 for experiments. We mark *CLIP*, *ConvNeXt*, *SAM*, *DINOv2*, *Pix2Struct*, and *EVA-02-L* as *CL*, *CN*, *SA*, *DI*, *PS*, and *EV*, respectively. A round-robin scheme, as shown in Fig. 4, is adopted. We first use the two top-performing vision encoders, *CLIP* and *ConvNeXt*, as the basis and gradually add one more vision encoder each time. In each round, the best-performing vision encoder combination is retained for the next round.

Fig. 4 reveals several insights. Generally, **introducing additional vision encoders enhances the performance**. This indicates that the distinct advantages of different encoders can be preserved and utilized; for example, integrating the *EVA-02* encoder improves metrics on the POPE benchmark. Although individual metrics may vary, the aggregated performance shows an upward trend, as evidenced by normalized average metrics, suggesting that the overall efficacy of the system is enhanced with more encoders. Also, Fig. 4 shows that the best combination of vision experts are *CLIP*, *ConvNeXt*, *SAM*, *Pix2Struct*, and *EVA-02*. We will use this recipe in the final model.

| #Encoder | Encoder Combination | Config | #Params (M) | FLOPs (G) | Img/Sec | Avg. |
|---|---|---|---|---|---|---|
| 2 | CL + CN | X2 | 1155.2 | 3347.2 | 18.1 | 681.5 |
| 3 | CL + CN + DI | | 1460.6 | 3659.9 | 15.1 | 685.4 |
| | CL + CN + SA | | 1463.9 | 4657.8 | 8.8 | 690.4 |
| | CL + CN + PS | | 1669.6 | 4373.2 | 6.9 | 685.1 |
| | CL + CN + EV | X3 | 1459.6 | 4280.9 | 9.1 | **690.7** |
| 4 | CL + CN + EV + DI | | 1765.1 | 4593.6 | 8.3 | 688.0 |
| | CL + CN + EV + SA | | 1768.4 | 5591.5 | 5.9 | 689.4 |
| | CL + CN + EV + PS | X4 | 1974.1 | 5306.9 | 5.0 | **694.6** |
| 5 | CL + CN + EV + PS + DI | | 2279.5 | 5619.5 | 4.7 | 684.7 |
| | CL + CN + EV + PS + SA | X5 | 2282.8 | 6617.4 | 3.8 | **697.1** |
| 6 | CL + CN + EV + PS + SA + DI | X6 | 2588.2 | 6930.1 | 3.6 | 686.8 |

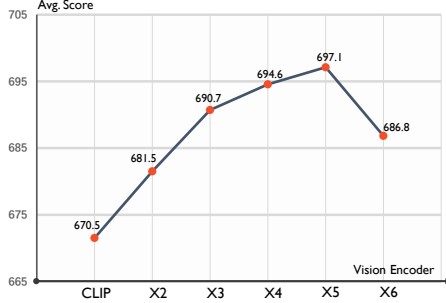

Figure 4: **Results of vision expert selection process.** *CL*, *CN*, *EV*, *PS*, *SA* and *DI* denote *CLIP*, *ConvNeXt*, *EVA-02*, *Pix2Struct*, *SAM* and *DINOv2*, respectively. (Left) The performance of various vision encoder combinations is presented, highlighting how different pairings influence overall effectiveness. "#Parames", "FLOPs" and "Img/Sec" denote the model size, complexity and throughput (bs=4) of the vision encoder. (Right) The curve illustrates the average score as the number of vision encoders increases. Each point on the curve represents the best-performing combination for the corresponding number of vision encoders. No *Pre-Alignment* is used in this comparison.

## 3 EXPERIMENTS

In this section, we take the findings and the best-explored design from Section 2 and compare them against the current state-of-the-art MLLMs on different tasks.

### 3.1 IMPLEMENTATION DETAILS

**Language models.** We use Vicuna-v1.5-7B (Chiang et al., 2023), Llama3-8B (AI@Meta, 2024) and Vicuna-v1.5-13B (Chiang et al., 2023) as the language models.

**Vision encoders.** We follow the best X4 and X5 configurations, where the interpolated *CLIP-448* and pre-aligned vision experts are channel-concatenated, and trained following the exact best training recipes in Figure 3 and Table 5.

**Training recipe.** On Eagle1.8M, we follow the base recipe in Section 2.1 with encoder learning rate the same as SFT (2e-5). On Cambrian data, we follow Tong et al. (2024) with PT/SFT bs=1024.

### 3.2 MAIN RESULTS

**Evaluation on visual question answering tasks.** We compare *Eagle* model series across three Visual Question Answering (VQA) benchmarks, including GQA (Hudson & Manning, 2019), VQAv2 (Goyal et al., 2017) and VizWiz (Gurari et al., 2018). As shown in Table 6, *Eagle-X5* achieves state-of-the-art performance on GQA and VQAv2, underscoring the advantages with additional vision experts.

**Evaluation on OCR and chart understanding tasks.** To evaluate the OCR, document, and chart understanding capabilities of *Eagle*, we benchmark our model on OCRBench (Liu et al., 2023f), TextVQA (Singh et al., 2019), and ChartQA (Masry et al., 2022). As illustrated in Table 6, our model significantly surpasses competitors on TextVQA, benefiting from its high-resolution architecture and integration of different vision encoders. Notably, *Eagle* maintains a straightforward design, supporting up to 1024x1024 resolution without requiring complex tile decomposition of images.

Fig. 5 shows some examples of OCR and document understanding cases. With high-resolution adaptation and more vision experts, our model can identify small text within images and accurately extract information according to the users' instructions. To better understand the benefits of introducing experts pre-trained on other vision tasks, we visualize the results of a model with only the *ConvNeXt* and *CLIP* vision encoders, compared to the results of *Eagle-X5* in Fig. 5. With the full set of vision encoders, the model can successfully correct mistakes, showing that even when equipped with high-resolution vision encoders pre-trained on vision-language alignment, the model's abilities can still be enhanced by integrating additional vision experts pre-trained on diverse vision tasks.

Table 6: **Main results with base training data.** SQA$^I$ denotes image split of ScienceQA.

| | Model | MME | MMB | SEED | MathVista | MMMU | POPE | SQA$^I$ | GQA | VizWiz | VQAv2 | OCR | TextVQA | ChartQA |
|---|---|---|---|---|---|---|---|---|---|---|---|---|---|---|
| *Vicuna-7B & Qwen-7B* | LLaVA-1.5 (Liu et al., 2023c) | 1510 | 64.3 | 58.6 | - | - | 85.9 | 66.8 | 62.0* | 50.0 | 78.5* | 297 | 58.2 | - |
| | LLaVA-NeXt (Liu et al., 2024a) | 1519 | 67.4 | 70.2 | 34.6 | 35.8 | 86.5 | 70.1 | 64.2* | 57.6 | 80.0* | 490 | 64.9 | - |
| | InternVL (Chen et al., 2023f) | 1525 | - | 65.4 | - | - | 86.4 | - | 62.9* | 52.5 | 79.3* | - | 57.0 | - |
| | LLaVA-HR (Luo et al., 2024) | 1554 | - | 64.2 | - | - | 87.6 | 65.1 | 64.2* | 48.7 | 81.9* | - | 67.1 | - |
| | Monkey (Li et al., 2024c) | - | - | - | - | - | - | - | 60.7* | **61.2*** | 80.3* | 514 | 67.6 | 65.1 |
| | Mini-Gemini (Li et al., 2024b) | 1523 | 65.8 | - | 32.2 | 36.8 | - | 71.1 | 64.5* | - | - | 477 | 65.2 | - |
| | Eagle-X5 | 1528 | 68.4 | **73.9** | 37.0 | 36.3 | **88.8** | 70.0 | **64.9*** | 54.4 | 83.4* | 529 | 71.2 | 67.7 |
| | Eagle-X5 (+Pre-Align) | **1582** | **69.7** | 73.7 | **38.2** | **38.0** | 88.7 | **71.9** | 64.6* | 58.7 | **83.6*** | **566** | 71.9 | **69.3** |
| *Vicuna-13B* | LLaVA-1.5 (Liu et al., 2023c) | 1531 | 67.7 | 61.6 | - | 36.4 | 85.9 | 71.6 | 63.3* | 53.6 | 80.0* | 331 | 61.3 | - |
| | LLaVA-NeXt (Liu et al., 2024a) | 1575 | 70.0 | 71.9 | 35.3 | 36.2 | 86.2 | **73.5** | 65.4* | 60.5 | 82.8* | 514 | 67.1 | 62.2 |
| | InternVL (Chen et al., 2023f) | 1546 | - | - | - | - | 87.1 | - | 63.9* | 54.6 | 80.2* | 517 | 58.7 | - |
| | LLaVA-UHD (Xu et al., 2024) | 1535 | 68.0 | - | - | - | 89.1 | 72.0 | 65.2* | 56.1 | 81.7* | - | 67.7 | - |
| | LLaVA-HR (Luo et al., 2024) | 1540 | - | 64.5 | - | - | 87.8 | 68.1 | 64.8* | 57.9 | 82.6* | - | 68.1 | - |
| | Mini-Gemini (Li et al., 2024b) | 1565 | 68.6 | 70.6 | 37.0 | 37.3 | - | 71.9 | 65.8* | - | - | 466 | 65.9 | 56.6 |
| | Eagle-X5 | **1609** | 69.2 | 74.1 | 38.8 | 36.6 | 87.8 | 72.7 | **66.2*** | 59.3 | 83.8* | 574 | **74.2** | 69.9 |
| | Eagle-X5 (+Pre-Align) | 1605 | **71.6** | **74.9** | **42.7** | **38.5** | **89.2** | **75.5** | 64.6* | **60.9** | **84.5*** | **598** | 73.3 | **72.1** |

**Evaluation on multimodal benchmarks.** We evaluate *Eagle* on seven benchmarks for MLLMs to demonstrate its capabilities from different perspectives, including MME (Fu et al., 2023), MM-Bench (Liu et al., 2023e), SEED (Li et al., 2023c), MathVista (Lu et al., 2024), MMMU (Yue et al., 2024), ScienceQA (Saikh et al., 2022), and POPE (Li et al., 2023e). Specifically, MME, MMBench, and SEED assess the overall performance on various real-world tasks based on reasoning, recognition, knowledge, and OCR. MMMU focuses on challenging problems from diverse domains that require college-level knowledge. POPE evaluates the visual hallucinations of MLLMs. The metrics used in our paper adhere to the default settings of these benchmarks. We report the perception score for MME, the `en_dev` split for MMBench, the `image` split of SEED, the `test-mini` split of MathVista, the `val` split of MMMU, the F1-score of POPE, and the `image` split of SQA to align with the reported scores from other models.

From the data presented in Table 6, *Eagle* consistently surpasses existing models across various MLLMs on SEED and MME, demonstrating the comprehensive knowledge and reasoning abilities of *Eagle*. With the help of vision encoders on object-centric tasks, *Eagle* also achieves the best performance on the POPE benchmark. Additionally, the *Pre-Alignment* strategy discussed in Sec. 2.5 has been found to further enhance performance when integrating multiple task-specific vision backbones.

Table 7: **Comparison between different training strategies.** "1 epoch" means we train *Eagle* for 1 epoch in the supervised fine-tuning stage. 'unlock*' means we unlock vision encoders in the pre-training stage. More details in Table. 13.

| Config Summary | Pre-align | Pre-train | Fine-tune | Avg. |
|---|---|---|---|---|
| 1 epoch | ✗ | LLaVA-595K | Eagle1.8M | 697.1 |
| 2 epoch | ✗ | LLaVA-595K | Eagle1.8M | 698.3 |
| 1 epoch, unlock* | ✗ | LLaVA-595K | Eagle1.8M | 698.0 |
| 1 epoch, unlock* | ✗ | LLaVA-595K+Eagle1.8M | Eagle1.8M | **699.5** |
| 1 epoch | Eagle1.8M | LLaVA-595K | Eagle1.8M | 706.6 |
| 1 epoch, unlock* | Eagle1.8M | LLaVA-595K | Eagle1.8M | 707.1 |
| 1 epoch, unlock* | LLaVA-595K+Eagle1.8M | LLaVA-595K | Eagle1.8M | 707.8 |
| 1 epoch, unlock* | LLaVA-595K+Eagle1.8M | LLaVA-595K+Eagle1.8M | Eagle1.8M | **708.9** |

This approach not only mitigates the inherent biases of each vision expert and the synergy between different modalities but also establishes a robust framework for multiple-expert fusion.

**Study on more advanced training recipes.** Table. 7 presents our step-by-step experiments to study the training recipes. We found that the best recipe is to first pre-align each vision expert on *LLaVA-595K + Eagle1.8M*. In the pretraining stage, we combine all vision experts from the first step and training both the projector and vision experts on *LLaVA-595K + Eagle1.8M*. Finally, we train the whole model on the *Eagle1.8M*.

**Comparison with *Cambrian-1*.** Using the same pre-training and supervised fine-tuning datasets from *Cambrian-1* (Tong et al., 2024), *Eagle* demonstrates superior performance across all the evaluated

Table 8: **Results using the same training data as *Cambrian-1*** (Tong et al., 2024). SQA[I] denotes ScienceQA-IMG (Saikh et al., 2022). RWQA denotes the RealworldQA (xAI, 2024).

| Model | Knowledge | | | | | General | | | | | OCR and Chart | | | | | Vision-Centric | | |
|---|---|---|---|---|---|---|---|---|---|---|---|---|---|---|---|---|---|---|
| | Avg | SQA[I] | MMMU | MathVista | AI2D | Avg | MME | MMB | SEED | GQA | Avg | ChartQA | OCR | TextVQA | DocVQA | Avg | MMVP | RWQA |
| *Llama3-8B* | | | | | | | | | | | | | | | | | | |
| MGM-HD | 55.7 | 75.1 | 37.3 | 37.0 | 73.5 | 72.7 | **1606** | 72.7 | 73.2 | 64.5 | 62.9 | 59.1 | 47.7 | 70.2 | 74.6 | 40.4 | 18.7 | 62.1 |
| Cambrian-1 | 61.3 | 80.4 | 42.7 | 49.0 | 73.0 | 73.1 | 1547 | **75.9** | 74.7 | 64.6 | 71.3 | 73.3 | 62.4 | 71.7 | 77.8 | 57.6 | 51.3 | 64.2 |
| Eagle-X5 | **65.2** | **84.1** | **43.5** | **56.9** | **76.2** | **74.0** | 1587 | 75.5 | **76.5** | **64.9** | **77.0** | **80.7** | **62.6** | **76.7** | **87.1** | **59.6** | **52.0** | **67.2** |
| *Vicuna-13B* | | | | | | | | | | | | | | | | | | |
| MGM-HD | 54.1 | 71.9 | 37.3 | 37.0 | 70.1 | 70.7 | 1597 | 68.6 | 70.6 | 63.7 | 60.8 | 56.6 | 46.6 | 70.2 | 69.8 | 38.4 | 19.3 | 57.5 |
| Cambrian-1 | 60.2 | 79.3 | 40.0 | 48.0 | 73.6 | 73.7 | 1610 | **75.7** | 74.4 | 64.3 | 71.3 | 73.8 | 61.9 | 72.8 | 76.8 | 52.2 | 41.3 | 63.0 |
| Eagle-X5 | **63.8** | **82.6** | **42.2** | **54.6** | **73.8** | **74.6** | **1651** | **75.7** | **75.0** | **65.0** | **75.7** | **78.6** | **62.4** | **74.9** | **86.7** | **54.8** | **44.6** | **65.0** |
| *Yi-34B* | | | | | | | | | | | | | | | | | | |
| MGM-HD | 62.4 | 77.7 | 48.0 | 43.4 | 80.5 | 76.2 | 1659 | 80.6 | 75.3 | 65.8 | 68.1 | 67.6 | 51.8 | 74.1 | 78.9 | 52.3 | 37.3 | 67.2 |
| Cambrian-1 | 67.0 | **85.6** | 49.7 | 53.2 | **79.7** | **76.8** | **1689** | **81.4** | 75.3 | **65.8** | 71.9 | 75.6 | 60.0 | 76.7 | 75.5 | **60.3** | **52.7** | 67.8 |
| Eagle-X5 | **68.6** | 85.5 | **53.2** | **57.9** | 79.1 | 76.3 | 1677 | 81.0 | **75.6** | 64.9 | **75.4** | **77.2** | **62.4** | **78.8** | **83.0** | 59.8 | 50.0 | **69.5** |

benchmarks without bells and whistles. As shown in Table 8, *Eagle* outperforms the *Cambrian-1* counterparts considerably for the *OCR and Chart* category. Consistent improvements are also observed for the *General*, *Knowledge*, and *Vision-Centric* categories, showing the robustness and generalization ability of the improved perception design in *Eagle*.

# 4  RELATED WORK

**Multimodal large language models.** Our work is related to the general architecture design of multimodal large language models. Besides the line of representative open-source research mentioned in the introduction section, other notable families of MLLMs include, but are not limited to *MiniGPT-4* (Zhu et al., 2023; Chen et al., 2023a), *Lynx* (Zeng et al., 2023), *Otter* (Li et al., 2023b;a), *Qwen-VL* (Bai et al., 2023), *CogVLM* (Wang et al., 2023b; Hong et al., 2024), *VILA* (Lin et al., 2023a), *GPT-4V* (Achiam et al., 2023), *Gemini* (Team et al., 2023), and *Llama 3.1* (Dubey et al., 2024). Depending on how vision signals are integrated into the language model, MLLMs can be broadly categorized into "cross-modal attention" ones and "prefix-tuning" ones (Yin et al., 2024). The former injects visual information into different layers of LLMs using cross-modal attention (Alayrac et al., 2022; Li et al., 2023b), whereas the latter views the visual tokens as part of the language token sequence and directly append them with text embeddings (Liu et al., 2023d; Li et al., 2022; 2023d). Our model belongs to the prefix-tuning family by following a *LLaVA*-styled multimodal architecture. Considering that MLLM is a fast-growing field, readers are recommended to refer to more detailed studies and surveys such as Yin et al. (2024); Li et al. (2024a); Fei et al. (2024).

**Vision encoder design for MLLMs.** Our work is closely related to research focusing on improved vision encoder designs for MLLM. Early works (Liu et al., 2023d; Li et al., 2022) usually adopted vision encoder pre-trained on vision-language alignment tasks such as *CLIP* (Radford et al., 2021) and *EVA-CLIP* (Sun et al., 2023). Stronger vision encoders such as *SigLIP* (Zhai et al., 2023) and *InternVL* (Chen et al., 2023f) have been proposed to improve vision-language tasks with improved designs, larger model sizes, and better training recipes. Considering that the models are often pre-trained on low-resolution images and lack the ability to encode fine-grained details, higher resolution adaptation is often performed to increase the MLLM input resolution (Chen et al., 2023c;d; Beyer et al., 2024; Chen et al., 2024).

Besides higher resolution adaptation, models such as *LLaVA-NeXT* (Liu et al., 2024a), *LLaVA-UHD* (Xu et al., 2024), *Monkey* (Li et al., 2024c), *InternLM-XComposer* (Dong et al., 2024), and *InternVL* (Chen et al., 2024) use adaptive tiling to divide high-resolution input into lower resolution patches that are processed separately. Although the ability to handle higher resolution is similarly made possible with the introduction of additional vision experts, its spirit is slightly orthogonal to the tiling techniques in terms of diversifying the visual representations. Both techniques are compatible and can be combined together.

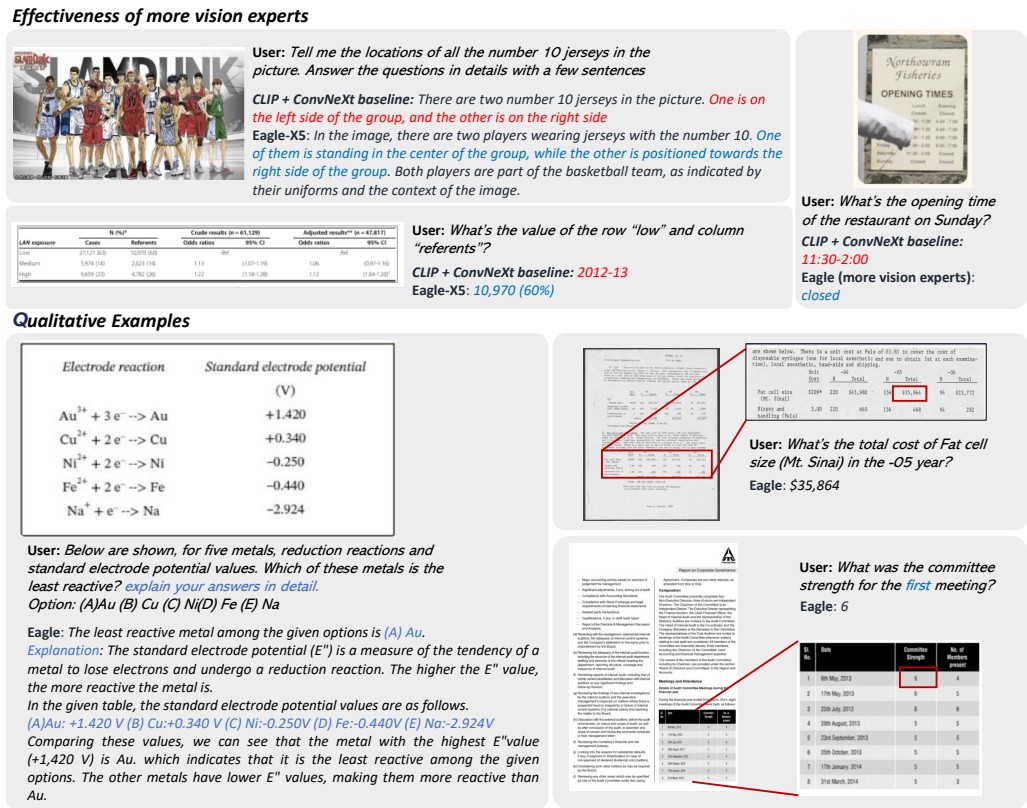

Figure 5: **Qualitative comparison of different numbers of vision experts.** *Baseline* means *Eagle* with only *CLIP+ConvNext*. **More Vision Experts** denotes the *Eagle-X5* model. We highlight a failure case in RED. BLUE indicates the correct answers. With more vision experts, *Eagle* can more precisely capture the information in the image and provide the correct answer.

Our work is most related to existing models using multiple vision encoders for improved perception. *Mini-Gemini* (Li et al., 2024b) and *LLaVA-HR* (Luo et al., 2024) propose to fuse high-resolution visual features into the low-resolution visual tokens. Apart from the resolution issue, these pre-trained vision encoders may lack specific abilities such as reading text and localizing objects. Hence, a series of works have integrated vision models pre-trained on different tasks for more comprehensive capabilities. For example, *Mousi* (Fan et al., 2024), *Prismatic VLM* (Karamcheti et al., 2024) and *Brave* (Kar et al., 2024) fuse visual tokens from different vision encoders by concatenating along the channel or token direction. There are also more complex approaches, including knowledge distillation Ranzinger et al. (2024), augmenting the input prompt with the information output by the vision experts (Lee et al., 2024; He et al., 2024; Liu et al., 2024b), or using a routing network to assign input to proper vision experts (Zong et al., 2024). In particular, *Prismatic VLM* (Karamcheti et al., 2024) systematically explores the design space of MLLMs across various dimensions, including data, training recipes, and notably. It also includes the vision encoder ensemble as part of its enhancement strategies. However, it lacks a comprehensive ablation study and discussion addressing the challenges associated with combining multiple vision encoders.

## 5 CONCLUSION

We conduct an in-depth analysis study on the design space for integrating vision encoders for multimodal large language models. Unlike previous works that focus on designing novel fusion paradigms, we find systematic design choice matters and discover a series of useful techniques. Step by step, we optimize the training recipes of individual vision encoders, identify an extendable and efficient fusion method, and gradually combine vision encoders with different domain knowledge. The results show the importance of basic design space. We hope our work can serve as a new basis and bring new inspiration for the vision encoder design for MLLM.

## 6 ACKNOWLEDGMENTS

The team would like to thank Yunhao Fang and Jason Lu for sharing the data and training recipes, and Yawen Luo for the assistance on figure editing. We thank Wei Ping, Zhuolin Yang, Wenliang Dai, Nayeon Lee, Boxin Wang, Ilia Karmanov, Lukas Voegtle, Philipp Fischer, Matthieu Le and Tuomas Rintamaki for their assistance on the internal codebases. We also thank the valuable discussions and input from Zhiqi Li, Guo Chen, Shilong Liu, Jihao Liu, Ming-Chang Chiu, Yunze Man, Shiyi Lan, Nadine Chang, Maying Shen, Vibashan VS, Jenny Schmalfuss, Jose Alvarez, Amala Sanjay Deshmukh, Mike Ranzinger, Greg Heinrich, Pavlo Molchanov, Vidya Murali, Parthasarathy Sriram, Mohammad Shoeybi, Song Han, Ofri Masad, Osvald Nitski, Qing Miao, Yao Xu, Jane Scowcroft, Dmitry Chichkov and Padmavathy Subramanian. Finally, the team would like to thank the Hugging Face Team that for the support of ZERO GPU demo, and the NVIDIA infrastructure team for their prompt and helpful assistance. Min Shi is partly supported by NSF Award #2427478 and #2229873.

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

# A APPENDIX

## A.1 BENCHMARK DETAILS

In this section, we provide the additional benchmark details for the tables in Section 2. The detailed comparison of different adaptation methods for the CLIP encoder are shown in Table 9. Table 10 shows the comparison between different vision encoders on all the adopted benchmarks. Table 11 list the detailed results on the vision encoder fusion methods. Table 12 shows the comparison between different vision encoder combinations on each benchmark.

Table 9: **Comparison of different high-resolution adaption methods to strengthen CLIP model (336x336).** RWQA denotes the RealworldQA (xAI, 2024).

| Vision Encoder | Unfreeze | Res | GQA | VizWiz | MME | SEED | OCR | DocVQA | ChartQA | AI2D | POPE | RWQA | SQA | Avg. |
|---|---|---|---|---|---|---|---|---|---|---|---|---|---|---|
| *Original* | ✗ | 336 | 63.2 | 55.1 | 1574 | 70.2 | 354 | 59.6 | 42.1 | 71.1 | 86.7 | 58.0 | 72.5 | 616.5 |
| *Original* | ✓ | 336 | 60.9 | 54.0 | 1501 | 61.8 | 305 | 56.4 | 24.5 | 69.3 | 80.9 | 53.6 | 72.0 | 562.6 |
| *Interpolate* | ✗ | 448 | 62.7 | 53.8 | 1470 | 69.1 | 299 | 58.4 | 37.7 | 71.2 | 85.7 | 55.3 | 69.6 | 589.7 |
| *Interpolate* | ✓ | 448 | 65.6 | 57.8 | 1534 | 73.7 | 526 | 65.0 | 61.1 | 73.7 | 87.3 | 57.7 | 71.5 | 670.5 |
| *Interpolate* | ✓ | 672 | 64.9 | 55.7 | 1503 | 72.4 | 509 | 64.6 | 62.0 | 72.2 | 87.1 | 57.4 | 71.2 | 674.2 |
| *Tiled-input* | ✓ | 672 | 63.0 | 54.9 | 1529 | 72.5 | 435 | 64.9 | 65.7 | 71.5 | 87.6 | 57.0 | 71.4 | 673.9 |
| *InternVL* | ✗ | 448 | 63.6 | 56.9 | 1537 | 71.7 | 529 | 65.0 | 58.6 | 72.9 | 87.4 | 59.2 | 70.2 | 661.9 |
| *InternVL* | ✓ | 448 | 65.6 | 57.8 | 1534 | 73.7 | 526 | 65.0 | 61.1 | 73.7 | 87.3 | 58.8 | 71.5 | 671.5 |

Table 10: **Comparison between different vision experts as the MLLM encoders.**

| Category | Vision Encoder | Unfreeze | Res | GQA | VizWiz | MME | SEED | OCR | DocVQA | ChartQA | AI2D | POPE | RWQA | SQA | Avg. |
|---|---|---|---|---|---|---|---|---|---|---|---|---|---|---|---|
| *VL Alignment* | *ConvNeXt* | ✗ | 1024 | 63.3 | 53.5 | 1526 | 70.6 | 404 | 70.4 | 60.8 | 71.6 | 87.5 | 57.1 | 68.6 | 635.0 |
| | | ✓ | 1024 | 63.3 | 54.4 | 1510 | 72.9 | 518 | 77.9 | 67.0 | 72.1 | 88.1 | 58.8 | 68.6 | 659.7 |
| *Segmentation* | *SAM* | ✗ | 1024 | 57.3 | 49.0 | 1216 | 56.9 | 38 | 20.1 | 17.4 | 69.2 | 84.3 | 49.2 | 66.8 | 471.1 |
| | | ✓ | 1024 | 60.2 | 51.5 | 1291 | 65.9 | 35 | 21.2 | 17.8 | 70.7 | 86.4 | 54.1 | 65.7 | 494.7 |
| *Object Detection* | *EVA-02* | ✗ | 1024 | 63.1 | 51.1 | 1359 | 69.2 | 123 | 25.6 | 25.2 | 71.2 | 88.5 | 57.9 | 66.1 | 523.5 |
| | | ✓ | 1024 | 64.3 | 55.5 | 1449 | 72.7 | 358 | 57.1 | 57.5 | 72.2 | 88.3 | 59.6 | 67.7 | 614.4 |
| *Text Recognition* | *Pix2Struct* | ✗ | 1024 | 53.1 | 48.1 | 1296 | 53.4 | 460 | 71.0 | 61.0 | 69.6 | 79.2 | 46.7 | 65.5 | 578.7 |
| | | ✓ | 1024 | 54.9 | 47.3 | 1262 | 55.1 | 472 | 72.5 | 62.0 | 68.7 | 80.0 | 49.3 | 66.6 | 584.8 |
| *Self-Supervised* | *DINOv2* | ✗ | 448 | 62.4 | 53.1 | 1438 | 67.4 | 41 | 20.2 | 17.3 | 70.7 | 85.3 | 53.3 | 67.1 | 503.1 |
| | | ✓ | 448 | 64.2 | 55.5 | 1466 | 71.8 | 45 | 20.4 | 17.5 | 71.4 | 87.4 | 57.3 | 67.8 | 518.0 |

Table 11: **Comparison of different fusion methods for different vision experts.** "#Tokens(V)" denotes the number of visual tokens.

| Vision Encoders | Fusion | #Tokens(V) | GQA | VizWiz | MME | SEED | OCR | DocVQA | ChartQA | AI2D | POPE | RWQA | SQA | Avg. |
|---|---|---|---|---|---|---|---|---|---|---|---|---|---|---|
| | *Seq. Append* | 2048 | 64.8 | 54.5 | 1563 | 73.4 | 532 | 77.7 | 67.6 | 72.4 | 87.9 | 61.2 | 68.8 | 690.5 |
| | *Channel Concat.* | 1024 | 63.2 | 48.0 | 1497 | 73.5 | 551 | 77.7 | 67.0 | 72.4 | 88.3 | 59.1 | 70.7 | 681.5 |
| *CLIP + ConvNeXt* | *LLaVA-HR* | 1024 | 64.5 | 57.2 | 1538 | 72.0 | 498 | 74.5 | 63.8 | 72.3 | 87.7 | 59.2 | 68.7 | 678.7 |
| | *Mini-Gemini* | 1024 | 65.3 | 56.9 | 1548 | 72.9 | 478 | 68.3 | 63.2 | 71.5 | 87.3 | 59.7 | 69.4 | 672.5 |
| | *Deformable Attn.* | 1024 | 64.0 | 57.3 | 1504 | 72.7 | 463 | 69.5 | 64.4 | 73.3 | 87.4 | 62.8 | 68.9 | 674.3 |
| *CLIP + ConvNeXt* | *Seq. Append* | 3072 | 64.3 | 53.6 | 1539 | 73.2 | 525 | 77.9 | 67.0 | 72.3 | 87.4 | 60.1 | 69.5 | 686.2 |
| *+ SAM* | *Channel Concat.* | 1024 | 63.3 | 55.9 | 1528 | 73.3 | 545 | 78.9 | 67.2 | 72.3 | 88.4 | 59.2 | 70.0 | 690.4 |

Table 12: **Detailed comparison on vision encoder combinations.**

| #Encoder | Encoder Combination | GQA | VizWiz | MME | SEED | OCR | DocVQA | ChartQA | AI2D | POPE | RWQA | SQA | Avg. |
|---|---|---|---|---|---|---|---|---|---|---|---|---|---|
| 2 | CL + CN | 63.2 | 48.0 | 1497.0 | 73.5 | 551.0 | 77.7 | 67.0 | 72.4 | 88.3 | 59.1 | 70.7 | 681.5 |
| 3 | CL + CN + DI | 63.3 | 55.9 | 1528.0 | 73.3 | 545.0 | 78.9 | 67.2 | 72.3 | 88.4 | 59.2 | 70.0 | 690.4 |
| | CL + CN + SA | 64.6 | 55.3 | 1504.0 | 73.3 | 526.0 | 75.7 | 64.9 | 72.1 | 88.3 | 61.1 | 70.9 | 685.4 |
| | CL + CN + PS | 63.2 | 51.4 | 1497.0 | 73.3 | 550.0 | 78.5 | 65.9 | 73.1 | 87.7 | 60.3 | 70.5 | 685.1 |
| | CL + CN + EV | 63.2 | 51.7 | 1565.0 | 73.9 | 538.0 | 77.7 | 67.8 | 73.6 | 89.0 | 61.4 | 69.4 | 690.7 |
| | CL + CN + EV + DI | 63.6 | 54.9 | 1512.0 | 73.8 | 547.0 | 77.0 | 66.7 | 73.1 | 88.9 | 60.4 | 69.7 | 689.4 |
| | CL + CN + EV + SA | 64.3 | 57.7 | 1533.0 | 73.7 | 521.0 | 75.2 | 65.3 | 72.2 | 88.5 | 61.1 | 70.2 | 688.0 |
| | CL + CN + EV + PS | 64.8 | 56.5 | 1561.0 | 73.4 | 540.0 | 78.8 | 67.5 | 72.2 | 88.4 | 59.9 | 70.5 | 694.6 |
| 5 | CL + CN + EV + PS + SA | 64.7 | 59.1 | 1528.0 | 73.9 | 529.0 | 78.6 | 67.8 | 72.9 | 88.8 | 62.2 | 69.7 | 697.1 |
| | CL + CN + EV + PS + DI | 64.7 | 54.1 | 1506.0 | 73.7 | 541.0 | 75.1 | 64.9 | 72.7 | 88.3 | 60.0 | 70.3 | 684.7 |
| 6 | CL + CN + EV + PS + SA + DI | 63.8 | 57.8 | 1512.0 | 73.5 | 525.0 | 75.1 | 65.8 | 71.8 | 88.4 | 61.4 | 69.9 | 686.8 |

Table 13: **Comparison between different training strategies.** '1 epoch" means we train *Eagle* for 1 epoch in the supervised fine-tuning stage. 'unlock*" means we unlock vision encoders in the pre-training stage.

| Config | Prealign | Pretrain | Finetune | GQA | MME | OCR | SciQA | POPE | DocVQA | ChartQA | SEED | Vizwiz | AI2D | RWQA | Avg. |
|---|---|---|---|---|---|---|---|---|---|---|---|---|---|---|---|
| 1 epoch | ✗ | llava595k | Eagle1.8M | 64.7 | 1528 | 52.9 | 69.7 | 88.8 | 78.6 | 67.7 | 73.9 | 59.1 | 72.8 | 62.2 | 697.1 |
| 2 epoch | ✗ | llava595k | Eagle1.8M | 65.4 | 1539 | 51.4 | 70.3 | 87.9 | 79.8 | 67.9 | 73.8 | 58.5 | 73.5 | 62.7 | 698.3 |
| 1 epoch, unlock* | ✗ | llava595k | Eagle1.8M | 64.1 | 1541 | 54.4 | 71.5 | 88.5 | 79.1 | 68.5 | 74.0 | 56.6 | 72.2 | 61.9 | 698.0 |
| 1 epoch, unlock* | ✗ | llava595k+Eagle1.8M | Eagle1.8M | 65.3 | 1545 | 54.8 | 70.5 | 88.5 | 78.8 | 68.4 | 73.5 | 57.5 | 72.0 | 62.9 | **699.5** |
| 1 epoch | Eagle1.8M | llava595k | Eagle1.8M | 64.5 | 1579 | 55.4 | 71.6 | 88.7 | 79.3 | 68.9 | 73.9 | 58.9 | 73.9 | 63.4 | 706.6 |
| 1 epoch, unlock* | Eagle1.8M | llava595k | Eagle1.8M | 65.1 | 1561 | 56.0 | 71.3 | 88.5 | 79.3 | 69.5 | 74.2 | 59 | 73.5 | 63.4 | 707.1 |
| 1 epoch, unlock* | llava595k+Eagle1.8M | llava595k | Eagle1.8M | 64.5 | 1570 | 56.0 | 71.7 | 89 | 79.7 | 68.9 | 74.0 | 58.5 | 74.0 | 63.7 | 707.8 |
| 1 epoch, unlock* | llava595k+Eagle1.8M | llava595k+Eagle1.8M | Eagle1.8M | 64.6 | 1582 | 56.6 | 71.9 | 88.7 | 79.8 | 69.3 | 73.7 | 58.7 | 73.5 | 63.9 | **708.9** |

## A.2 ADDITIONAL QUALITATIVE RESULTS

Fig. 6 demonstrates that our model can perform reasoning based on the knowledge to answer some difficult questions. Fig. 7 shows some examples of OCR and document understanding cases.

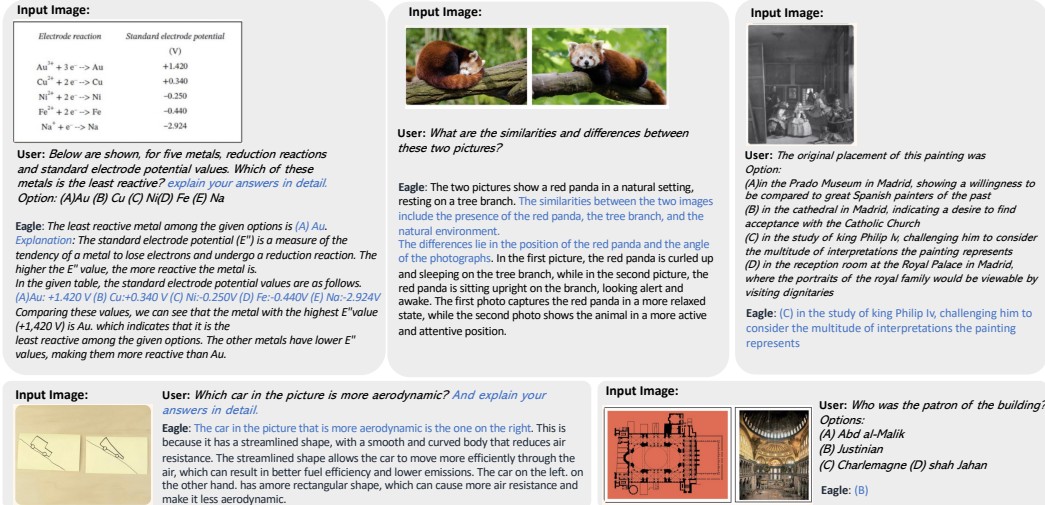

Figure 6: **Qualitative results of Eagle** on VQA tasks requiring college-level knowledge or deliberate reasoning.

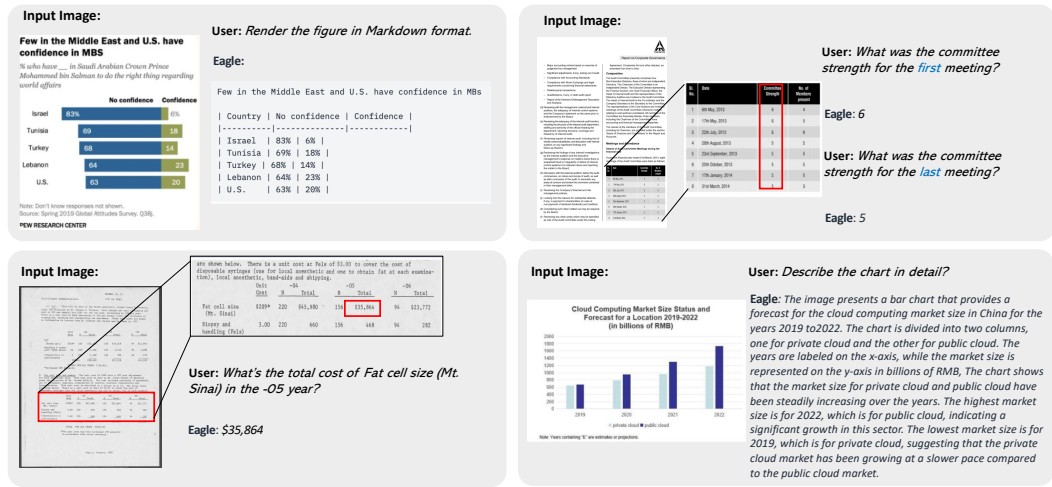

Figure 7: **Qualitative samples on OCR and document understanding tasks**. Eagle is able to extract useful information from small text.

