# OpenReview forum: "Eagle: Exploring The Design Space for Multimodal LLMs with Mixture of Encoders"
_ICLR.cc/2025/Conference — ICLR 2025 Spotlight_

### Official Review · Reviewer_aaRq · 2024-10-24

**Soundness:** 3
**Presentation:** 4
**Contribution:** 3
**Rating:** 8
**Confidence:** 4

**Summary:**

The paper explores strategies to inject information (visual tokens) from a mixture of pretrained visual encoders (experts) into LLM’s (Vicuna-12B) for finetuning MMLM’s. Namely, they explore how to incorporate: 1) High Resolution inputs which are needed for text understanding. 2) A number of visual encoders. Based on their ablation strategies, they propose the following training recipe.

* Simple linear interpolation for adaptation to high resolution.
* Unfreezing the image encoder instead of just the projection layer
* Channel concatenation to incorporate multiple vision encoder outputs.
* Prealigning the visual encoders with a LLM.
* Simple greedy strategy to incorporate multiple vision experts.

**Strengths:**

The paper is well written. They start with a base clip image encoder model and build towards the final recipe using careful ablations. They show that a simple recipe without any fancy tricks can lead to strong performance on a number of benchmarks. The paper also achieves strong performance on VLM benchmarks and is likely to have an impact on multimodal LLMs.

**Weaknesses:**

The clarity over some of the details in the paper can be expanded upon. See below for detailed feedback.

**Questions:**

The clarity over some of the details in the paper can be expanded upon.

* It’s a bit worrisome that the benchmark used to arrive at the recipe (Table 2-5 and Fig.4) is the same benchmark used in the final results (Table 6.). This would be great if authors confirm that the validation set is used for Table 2-5 and Fig.4 and the test-set for Table 6 or provide some evidence that the recipe is not overfitting to this particular benchmark.
* There is a final finetuning stage where the entire model, i.e LLM + projector + vision encoder is finetuned. In Table 7, it is mentioned that this is done on Eagle 1.8M, however on L141-L142, it is mentioned that this is done on multi-model conversation data. Further clarification on this will be helpful.
* Similarly, in Table 7 the best model is pretrained on LLava-595K + Eagle 1.8M. However, it seems Table 2-5 and Fig 4, the recipe is arrived at by only pretraining on Eagle 1.8M. Can the authors confirm this and clarify this in the paper?
* What is the effect of unfreezing vs freezing the image encoder on training time? Table 2 shows the number of flops are the same for both freezing and unfreezing. But while unfreezing one has to backpropagate through the encoder , so the flops should be different.
* What is the difference between 5th row of Table 5 (Unfreeze yes + prealign no + clip + convnext) and 2nd row of Table 4? In both cases, clip + convnext is used with channel concatenation and pre-alignment switched off. So why are the numbers different?
* The empirical comparison of flops between tiling and interpolation is provided in Table 2. I assume there is a tradeoff in the sense that tiling requires more forward passes but the self-attention might be less expensive because of the lesser number of tokens . So It would also be nice if some theoretical comparisons are also provided between interpolation and tiling with respect to number of tokens and resolution.
* I may have missed it but was the size of the Clip encoder L? And what resolution was the pretraining done on?

---

> ### Author Response · Authors · 2024-11-21
> **Response to Reviewer aaRq Part 1**
>
> __Detailed Benchmark Setting__
>
> We thank Reviewer aaRq for highlighting the benchmark setting. For Tables 2–5, we use the validation sets of VizWiz, SEED, DocVQA, and ChartQA for our investigations. For the final results, the models are evaluated on the test sets to ensure alignment with the evaluation protocols of other compared models.
> To minimize the risk of overfitting to specific benchmarks, we also reduce the overlap between the benchmarks used for our investigations and those used for final comparisons with other models.
>
> __Questions on Fine-tuning Data__
>
> Thank you for highlighting this potential confusion! On lines 141–142, our intention is to describe the general type and format of the fine-tuning data, which is multimodal conversations. For this purpose, we utilize Eagle 1.8M, a dataset we collected that includes a diverse mixture of multi-modal conversation data spanning various tasks.
>
> All the models from Table 2-5 and Fig. 4 are trained in two stages. On the first stage the model is pretrained on LLaVA-595k and the second stage on Eagle 1.8M. For experiments involving pre-alignment, as shown in Tables 6 and 8, the models undergo a three-stage training process. During the first pre-alignment stage and the second pretraining stage, we use a combination of LLaVA-595k and Eagle 1.8M as the training data.
>
> __Additional computation cost of unfreezing vision encoders__
>
> Thank you for the question! The throughputs and FLOPs are all tested under inference mode, which can not reflect the difference in back-propagation.
>
> To clarify the additional computational costs incurred during training, we have provided a detailed comparison of training time and memory usage in the table below. All experiments were conducted on 32 A100 GPUs across four nodes using DeepSpeed Zero-2 optimization. Each GPU processes a batch size of 4, resulting in a total batch size of 128.
>
> As shown in the table below, unfreezing the vision encoder during training increases both training time and memory usage. However, we find this step to be crucial for enhancing performance, particularly when the input resolution changes or when the vision encoder has not been pretrained on vision-language alignment tasks. For example, unfreezing the vision encoder during the second stage improves the average performance of EVA02 (pretrained on detection) from 543.7 to 639.1. Please refer to Table 3 for more detailed comparisons.
>
> Considering the performance gain and the fact that unfreezing the vision encoder during training does not increase inference time, we believe the performance gains fully justify the additional training time.
>
> | Model | Freeze Vision | Training Time | Memory |
> | -------- | ------- | ------- | ------- |
> | Eagle-X5-7B  | True  | ~16h  | 40GB   |
> | Eagle-X5-7B  | False  | ~24h  | 52GB   |
> | Eagle-X5-13B  | True  | ~23h  | 56GB   |
> | Eagle-X5-13B  | False  | ~33h  | 64GB   |
>
> __The comparison between tiling and interpolation__
>
> Thank you for the question! There is indeed a trade-off in FLOPs between the tiling and interpolation approaches. Tiling divides the input image into patches, which are processed by the ViT independently, whereas interpolation treats the image as a whole, allowing all patches to participate in the attention calculation. This is analogous to window attention versus global attention: tiling resembles window attention, while interpolation aligns with global attention.
>
> Hence, as shown in Table 2, under the same number of visual tokens and input resolutions, the tiling approach has fewer FLOPs. However, its throughput is lower, likely because throughput can be significantly influenced by hardware capabilities and computation parallelism. Advanced hardware and software optimizations, such as flash-attention, have greatly improved the efficiency of attention computation and helps boost parallel computation. Consequently, even when processing an input resolution of 672 directly, the system does not encounter a performance bottleneck. In contrast, tiling requires multiple forward passes (e.g., four for a single image), which reduces throughput due to limited parallelization opportunities.
>
> Theoretically, as with window and global attention, window attention has linear complexity with respect to the number of input tokens, whereas global attention has quadratic complexity. Hence, as the input resolution further scales up, it may hit the bandwidth of the platform and cause lower throughputs or even cause out-of-memory issues.
>
> __Question about Clip encoder L__
>
> The size of the CLIP-L encoder, as shown in table 2 has 304M parameters in total. It’s pretrained on 336 inputs by OpenAI. The pretrained checkpoint can be accessed at https://huggingface.co/openai/clip-vit-large-patch14-336.

---

> > ### Author Response · Authors · 2024-11-21
> > **Response to Reviewer aaRq Part 2**
> >
> > __Explanation of the 5th row of Table 5 and the 2nd row of Table 4__
> >
> > Thank you for this insightful question! The difference in numbers arises because the average score in Table 5 was initially calculated based on a different set of benchmarks than those used in Table 4. We have addressed this discrepancy by recalculating the values in Table 5 using the same benchmark set as in Table 4. The results align with our original findings. We have updated the manuscript accordingly to ensure consistency. We appreciate your attention to this detail, which helped us improve the clarity of our results.

---

### Official Review · Reviewer_GjPo · 2024-10-30

**Soundness:** 3
**Presentation:** 3
**Contribution:** 2
**Rating:** 6
**Confidence:** 4

**Summary:**

This paper conducts an empirical analysis study on the design space for integrating vision encoders for MLLMs. It optimizes the training recipes of individual vision encoders step by step, identify an extendable and efficient fusion method, and gradually combine vision encoders with different domain knowledge. Built upon these findings, this paper propose Eagle, the results somehow show the importance of basic design space.

**Strengths:**

Strength：

1. Extensive experiments demonstrate the importance of optimizing the design space of vision encoders.
2. The ablation studies are abundant.
3. This paper is well-organized and clearly-written.

**Weaknesses:**

Weakness:

1. The contribution is limited. PrismaticVLM[1] conducts similar empirical study on MLLMs, especially on the vision encoders. Although they only explored the combination of two encoders, they have achieved significant improvements while keeping a good efficiency. This paper extends the mixture of encoders up to six, but the performance only increases from 681.5 to 697 as shown in Fig.4, which is somehow trivial from my perspective. What's worse, it brings double parameters and slows down the inference by more than four times.
2. Missing related discussion. It is really strange there is not any discussion with PrismaticVLM, considering these two paper focus so closely.

[1] Prismatic VLMs: Investigating the Design Space of Visually-Conditioned Language Models. ICML 2024

**Questions:**

Please see the comments above.
My main concern is weakness 1, for such reasons i don't think this paper's contribution can lead to an acceptance.

---

> ### Author Response · Authors · 2024-11-21
> **Response to Reviewer GjPo Part 1**
>
> We thank Reviewer GjPo for kindly reminding us and apologize for missing this key paper. We agree that Prismatic VLM should be cited given its pioneering study in the design space of VLMs, including the use of a mixture of vision encoders and channel concatenation. __We have included detailed comparisons and discussions in the introduction, methodology, and related work sections in the revised manuscript (marked in orange)__. That being said, we would like to emphasize a few important points:
>
> __Strange not citing/discussing with PrismaticVLM__
>
> It is our mistake missing this important work. However, we did not try to avoid the discussion with related works, as the reviewer might have hinted here. Instead, we stated clearly in paper (Line 86) that "Our work is not the first one to leverage multiple vision encoders in MLLM." Many of the cited works are concurrent or even earlier than Prismatic VLM with MoVE (Mixture of Vision Encoders) designs. For instance, one of the cited works, SPHINX (Nov 13 2023), also uses channel concatenation to mix multiple encoders, similar to Prismatic VLM (Feb 12 2024). This work does not try to paint itself as the first one to propose MoVE or MoVE with channel concatenation. But we agree with the reviewer that it's important to make this part clearer.
>
> __Findings/contributions are trivial__
>
> We emphasize that this work is not a mere extension of Prismatic VLM to more vision encoders. The differences in contributions are significant between the two works:
>
> 1) Goals and exploration spaces are quite different. The purpose of Prismatic VLM aims to study the VLM design in general where MoVE is part of the explored designs bringing benefits, whereas this work delves much deeper into the design of MoVE by studying/comparing a spectrum of MoVE strategies carefully, in response to the lack of systematic study in MoVE. Our study covers critical aspects such as vision encoder selection, fusion paradigms, and pre-training strategies tailored for combining multiple vision encoders. These contributions introduce insights complementary to Prismatic VLM and provide novel, valuable knowledge to the community.
>
> 2) Key conclusions are different and more comprehensive. Prismatic VLM, like many other cited works in the paper, belongs to the family of works with frozen vision encoders. In fact, Prismatic VLM found that unlocking vision encoder during fine-tuning hurts the performance, whereas this work leads to a different conclusion that unlocking vision encoder during fine-tuning is the key to help MoVE work. This difference is probably caused by the improved data and training recipes adopted in this work, and better echoes the more recent findings in other works, such as Cambrian-1 and LLaVA-OneVision. In fact, our frozen vision encoder baselines can be seen as a proxy of the proposed strategy in Prismatic VLM under our apples-to-apples comparison. And the improvements from unlocking and pre-alignment are significant. As such, it may be incorrect to conclude that the contributions are limited.
>
> 3) Comparing the performance gain is "comparing apples with oranges". Compared to Prismatic VLM, the baseline of this work started with larger data (Eagle 1.8M), stronger baseline (e.g., higher TextVQA and AI2D) and covers a wider range/domain of benchmarks. Achieving the same delta in performance gain is thus significantly more difficult. On the other hand, a significant portion of gain in Prismatic VLM (e.g., Table 7 SigLIP + DINOv2) came from object-centric datasets such as RefCOCO/RefCOCOg/RefCOCO+/OCID-Ref, which is not that surprising considering the weak object localization of SigLIP and the strong object localization of DINOv2. Thus, directly comparing the two works in performance gain is not fair and does not help to draw meaningful conclusions. More importantly, we aim to demonstrate MoVE with more encoders as an effective way to scale-up the vision foundation of VLMs and benefit their performance. This is clear in comparison to InternVL which has 6B parameters but shows significantly lower results.

---

> ### Author Response · Authors · 2024-11-21
> **Response to Reviewer GjPo Part 2**
>
> 4) The additionally introduced costs in vision encoder is relatively marginal compared to the LLM auto-regressive generation as a whole. To provide quantitative evidence, we evaluate image captioning as a real-world use case to measure the inference speed of different vision encoders. The experiment uses Vicuna-7B as the base model, employing float16 precision and KV cache optimization for efficient LLM generation. The generation process is halted after producing 128 tokens (approximately 100 words). The input consists of a 1024 x 1024 resolution image and a prompt—“Describe this image in detail.”—which results in 50 input text tokens plus a chat template.
>
>    We calculate four distinct computational metrics:
>
>     - Generation time: The time taken to generate 128 tokens, measured from the input of the model to the output of the final token.
>
>     - Vision encoder latency: The time required for the vision encoder to process the input image and produce feature maps.
>
>     - FLOPs of the vision encoder: The total floating-point operations performed by the vision encoder.
>
>     - FLOPs of the first forward pass: The total operations required to compute the hidden states for both the input image and text embeddings.
>
>    These metrics allow us to provide a comprehensive analysis of computational costs and efficiency for the vision encoders within the context of a full MLLM pipeline.
>
>
> | Model    | Input Resolution | Parameters | Generation Time | Vision Tower Latency | Vision Tower FLOPs | First Forward FLOPs | Avg. |
>  | -------- | ------- | ------- | ------- | ------- | ------- | ------- | ------- |
>  | InternVL  | 448   | 5536M   | 4.01s   | 0.08s   | 5670   | 13037   | 671.5   |
>  | CLIP (X1) High-res  | 448   | 304M   | 3.85s   | 0.02s   | 422   | 7533   | 670.5   |
>  | CLIP + ConvNeXt (X2)  | 1024   | 1155M   | 3.92s   | 0.04s   | 3347   | 10714   | 681.5   |
>  | CLIP + ConvNeXt + EVA-02 (X3)  | 1024   | 1460M   | 3.97s   | 0.07s   | 4281   | 11650   | 690.7  |
>  | Eagle-X4-7B  | 1024   | 1974M   | 4.10s  | 0.17s   | 5307   | 12681   | 694.6   |
>  | Eagle-X5-7B  | 1024   | 2283M   | 4.19s   | 0.24s   | 6617   | 13995   | 697.1   |
>
>    __Avg.__ means the average score across a number of VLM benchmarks.
>
>    The results show that the latency of the vision encoder constitutes only 5.7% of the total generation time for the heaviest model, X5. This proportion decreases further with larger language models, such as 3% for a 13B model, and becomes even smaller in scenarios involving longer token generation, such as multi-round conversations or detailed explanations.
>
>
>    In terms of floating-point operations (FLOPs), the vision encoder accounts for less than 50% of the operations during the initial forward pass. Meanwhile, the auto-regressive generation process requires repeated computation for each subsequent token, significantly outweighing the vision encoder's contribution to overall computation.
>
>
>    We would also like to mention that our current implementation is based on a naive PyTorch setup, where each vision encoder requires stand-alone processing code and is computed sequentially. However, in practice, each vision encoder can be assigned to a separate GPU since they operate independently. In such a scenario, the inference time for the entire vision encoder ensemble would be largely determined by the slowest vision encoder. For Eagle-X5, this would reduce the total inference time to ~0.10s instead of 0.24s.

---

> > ### Comment · Reviewer_GjPo · 2024-11-23
> > **Response**
> >
> > Thanks for your reply, i have raised my rating from 5 to 6.

---

> > > ### Author Response · Authors · 2024-11-24
> > > **Thank You!**
> > >
> > > Thank you very much for the time and constructive feedback!

---

### Official Review · Reviewer_1gkn · 2024-11-03

**Soundness:** 4
**Presentation:** 4
**Contribution:** 4
**Rating:** 8
**Confidence:** 4

**Summary:**

This paper's findings reveal several underlying principles common to various existing strategies, leading to a streamlined yet effective design approach. The paper discovers that simply concatenating visual tokens from a set of complementary vision encoders is as effective as more complex mixing architectures or strategies. Additionally, the paper introduces Pre-Alignment to bridge the gap between vision-focused encoders and language tokens, enhancing model coherence. The resulting family of MLLMs, Eagle, surpasses other leading open-source models on major MLLM benchmarks.

**Strengths:**

1. This paper has confirmed through comprehensive and integrated experiments the necessity of turning on vision encoders during the SFT stage. At the same time, the paper has discovered and validated that straightforward channel concatenation stands out as a simple yet competitive fusion strategy, offering the best efficiency and performance.
2. This paper proposes a pre-alignment stage where non-text-aligned vision experts are individually fine-tuned with a frozen LLM before being trained together.
3. The illustrations in this paper are accurate and concise, and the language expression is very refined, making it easy for people to understand.

**Weaknesses:**

1. Some details of the methods are missing. How are the projectors of each expert in stage 1 of Figure 3 dealt with in stage 2?
2. Some corrections to details. In Table 4 at line 317 of the text, it looks like it should be referring to Figure 4. The same applies below.

**Questions:**

## Justification For Recommendation And Suggestions For Rebuttal：
- Justification For Recommendation：Reference to Paper Strengths.

## Additional Comments For Authors：
It would be better to add the results of some other models that also use many vision encoders for comparison in Table 6.

---

> ### Author Response · Authors · 2024-11-21
> **Response to Reviewer 1gkn**
>
> Thanks for the constructive feedback! We hope to clarify the confusion and answer the questions below. Feel free to let us know if you have any further questions!
>
> __Details of the projector__
>
> In stage-2, we initialize only the weights of the vision encoder from stage-1, i.e., the pre-alignment training. The projectors, however, are newly initialized in the second stage. Inspired by your feedback, we believe it is worth exploring an alternative approach: concatenating the projectors from stage-1 to form a new projector. If this modification proves effective, we will update the manuscript accordingly.
>
> __Incorrect Cross-reference__
>
> Thank you for pointing this out. We have addressed this concern and made the necessary adjustments in the revised manuscript.

---

> > ### Comment · Reviewer_1gkn · 2024-11-26
> >
> > This is a great piece of work, and I look forward to your future progress.

---

### Official Review · Reviewer_La1k · 2024-11-03

**Soundness:** 3
**Presentation:** 3
**Contribution:** 2
**Rating:** 6
**Confidence:** 4

**Summary:**

The authors present an extensive study of mixture of encoders, providing ablations and experiments on various design choices. The authors recommend interpolating for higher resolutions, having a higher number of vision encoder experts, concatenating tokens in a channel-wise manner, and also compare training strategies. The method is compared against a lot of recent advances in this space, and has competitive performance on a variety of benchmarks.

**Strengths:**

- The paper is well articulated, with various steps properly backed up with experiments, and compared against significant number of baselines. The results indicate that the approach performs better than other mixture of vision encoders works.
- The better performance of interpolation over tiling in the initial steps is an important contribution towards efficiency, and is much appreciated
- The paper addresses an important issue, with multiple strides in the space of multimodal LLMs, it is essential to incorporate the best set of changes that work well together, and this paper is a good step towards this goal.
- Well written paper, easy to read

**Weaknesses:**

I will combine weaknesses and questions here:

(1) My major primary concern is about the efficiency of the approach, and I will list it below as a two-part concern -

- (**Most Imp Concern**) Fig 4 mentions the various combinations tried in the mixture of vision encoders. As compared to the 2 encoder setting (X2), the next mixture (X3) leads to a slight improvements at the expense of 50% reduction in throughput, and going further, X5 leads to more than 4x reductions in throughput. While the results in terms of accuracy improve, as also suggested by Table 6, it comes at a significant compute expense. It would be important to understand the author's perspective on this agenda. Additionally, in Table 6, it would be useful to mention the throughput of other baseline methods, wherever possible. In my humble opinion, if the proposed method has a severely lower throughput (high inference cost), it should also be compared against other methods having comparable throughput when comparing accuracy metrics.
- The paper suggests that unfreezing leads to better performance, however, it also increases the number of trainable parameters. The authors should provide details on the training overhead that this change leads to, so as to better understand the efficacy of the slight improvement.

(2) The motivation behind introducing the Deformable Attention baseline is unclear and the comparisons made with it are limited. Additionally, to motivate channel concatenation further over sequence concatenation, quantitative comparisons of the implications of handling longer sequence lengths would be helpful.

**Questions:**

Please see the weaknesses section above.

---

> ### Author Response · Authors · 2024-11-21
> **Concerns Regarding Performance Gains with More Vision Encoders**
>
> Thank you for your valuable feedback on performance gain of including more vision encoders!
>
> The performance gain after X2, though less significant on numbers, remains consistent. More importantly, it demonstrates a greater advantage compared to simply scaling a single encoder (e.g., InternVL-6B). This is attributed to the heterogeneity among expert architectures, the diversity of pre-trained knowledge, and improved coverage of various domain-specific expertise.
>
> Here’s how we interpret the relatively reduced improvement:
> 1) ConvNeXt is indeed a powerful vision encoder with comprehensive capabilities, pre-trained on vision-language alignment tasks. Its convolutional nature complements the CLIP encoder effectively and supports high-resolution. Additionally, ConvNeXt’s large size (846M parameters) provides greater capacity, resulting in a boost to overall performance. In comparison, the other added vision encoders are more task-specific, primarily enhancing performance on benchmarks closely related to their pretraining tasks.
>
> 2) The benefit in the expansion of domain-specific expertise and capabilities should not be overlooked. This is particularly true for a generalist model, and the benefits here are sometimes implicit and not fully reflected by the scores on paper. In particular, many additional vision encoders are task-specific. For instance, Pix2Struct helps document and chart understanding, while EVA02 and SAM help object-centric, localization, and open-world recognition tasks. As shown in Table 12 of the appendix, from X3 to X5 where more specialized encoders like Pix2Struct and SAM are added, the improvements are more task-specific—most pronounced on datasets such as VizWiz, DocVQA, and ChartVQA. Although these experts can’t boost the average performance significantly as a strong vision language experts, they can help solving complex real-world tasks with special representations inherited from its own pretraining task. First part of Fig. 5 has listed some of the examples.

---

> ### Author Response · Authors · 2024-11-21
> **Efficiency Trade-off of Adding More Vision Encoders**
>
> Here we address all the concerns on the efficiency trade-off of adding more vision experts.
>
> We want to first clarify that, for a MLLM, the additionally introduced costs in vision encoder is relatively marginal compared to the LLM auto-regressive generation as a whole. Hence, all of the methods share similar generation time with the same language model and generation length. To provide quantitative evidence, we evaluate image captioning to measure the inference speed of different vision encoders. The experiment uses Vicuna-7B as the base model, employing float16 precision and KV cache optimization for efficient LLM generation. The generation process is stopped after producing 128 tokens (approximately 100 words). The input consists of a 1024 x 1024 resolution image and a prompt—“Describe this image in detail.”—which results in 50 input text tokens plus a chat template.
> We calculate four distinct computational metrics:
> + Generation time: The time taken to generate 128 tokens, measured from the input of the model to the output of the final token.
> + Vision encoder latency: The time required for the vision encoder to process the input image and produce feature maps.
> + FLOPs of the vision encoder: The total floating-point operations performed by the vision encoder.
> + FLOPs of the first forward pass: The total operations required to compute the hidden states for both the input image and text embeddings.
>
> |               Model           | Input Resolution | Parameters | Generation Time | Vision Tower Latency | Vision Tower FLOPs | First Forward FLOPs |  Avg. |
> |:-----------------------------:|:----------------:|:----------:|:---------------:|:--------------------:|:------------------:|:-------------------:|:-----:|
> |          Eagle-X5-7B          |       1024       |    2283M   |      4.19s      |         0.24s        |        6617        |        13995        | 697.1 |
> |          Eagle-X4-7B          |       1024       |    1974M   |      4.10s      |         0.17s        |        5307        |        12681        | 694.6 |
> | CLIP + ConvNeXt + EVA-02 (X3) |       1024       |    1460M   |      3.97s      |         0.07s        |        4281        |        11650        | 690.7 |
> |      CLIP + ConvNeXt (X2)     |       1024       |    1155M   |      3.92s      |         0.04s        |        3347        |        10714        | 681.5 |
> |           CLIP (X1)           |        448       |    304M    |      3.85s      |         0.02s        |         422        |         7533        | 670.5 |
> |            InternVL           |        448       |    5536M   |      4.01s      |         0.08s        |        5670        |        13037        | 671.5 |
>
> The results show that the latency of the vision encoder constitutes only 5.7% of the total generation time for the heaviest model, X5. This proportion decreases further with larger language models, such as 3% for a 13B model, and becomes even smaller in scenarios involving longer token generation, such as multi-round conversations or detailed explanations. In terms of floating-point operations (FLOPs), the vision encoder accounts for less than 50% of the operations during the initial forward pass. Meanwhile, the auto-regressive generation process requires repeated computation for each subsequent token, significantly outweighing the vision encoder's contribution to overall computation.
>
> The reviewer mentioned that methods with similar inference cost should be compared for a fair comparison, which we think is very reasonable. We choose the current largest vision encoder, InternVL 6B, as the vision encoder with the same training data and recipe. With similar FLOPs and over 2x parameters, InternVL-6B still falls behind our method.
>
> In addition, we would also like to mention that our current implementation is based on a naive PyTorch setup, where each vision encoder is computed sequentially. However, in practice, each vision encoder can be assigned to a separate GPU since they operate in parallel independently. In such a scenario, the inference time for the entire vision encoder ensemble would be largely determined by the slowest vision encoder.

---

> ### Author Response · Authors · 2024-11-21
> **Efficiency Analysis of Unfreezing the Vision Tower**
>
> Here we list the training time comparison between unfreezing/freezing the vision encoders during supervised fine-tuning. The training is conducted on 32 A100 GPUs across four nodes using DeepSpeed Zero-2 optimization. Each GPU processes a batch size of 4, resulting in a total batch size of 128.
>
> | Model        | Freeze Vision | Training Time | Memory |
> |--------------|---------------|---------------|--------|
> | Eagle-X5-7B  | True          | ~16h          | 40GB   |
> | Eagle-X5-7B  | False         | ~24h          | 52GB   |
> | Eagle-X5-13B | True          | ~23h          | 56GB   |
> | Eagle-X5-13B | False         | ~33h          | 64GB   |
>
> As shown in the table below, unfreezing the vision encoder during training increases both training time and memory usage. However, we find this step to be crucial for enhancing performance, particularly when the input resolution changes or when the vision encoder has not been pretrained on vision-language alignment tasks. For example, unfreezing the vision encoder during the second stage improves the average performance of EVA02 (pretrained on detection) from 543.7 to 639.1. Please refer to Table 3 for more detailed comparisons. Considering the performance gain and the fact that unfreezing the vision encoder during training does not increase inference time, we believe the performance gains fully justify the additional training time.

---

> ### Author Response · Authors · 2024-11-21
> **Motivation of Introducing Deformable Attention Baseline**
>
> We thank Reviewer La1lk for the thoughtful feedback! We will include additional discussion on the motivation and implementation of the deformable attention (DA)-based fusion approach. We incorporated DA as a fusion method for two main reasons:
>
> + DA is widely adopted in various vision tasks, achieving state-of-the-art performance while maintaining high efficiency.
> + DA shares conceptual similarities with the MiniGemini architecture. It can also be seamlessly integrated.
>
> First, DA has become a critical component in many cutting-edge approaches across a range of vision tasks, such as DINO for object detection, Grounding-DINO for visual grounding, and BEVFormer for autonomous driving. Given its proven effectiveness in these applications, we were motivated to explore its potential for enhancing the visual representation within MLLMs.
>
> Second, DA closely resembles the patch information mining architecture proposed by the important compared method, Mini-Gemini, where each pixel attends to neighboring pixels to facilitate information fusion. However, DA can dynamically predict the neighborhood, enabling larger and more flexible receptive fields while maintaining high efficiency.
>
> Initially, we did not anticipate that simple direct concatenation would already yield strong performance. As a result, we tested several architectures in parallel, with DA considered an enhancement of MiniGemini’s fusion method. Therefore, we included DA in our comparisons to provide a comprehensive evaluation.

---

> ### Author Response · Authors · 2024-11-21
> **Further Discussion on Channel Concatenation and Sequence Appending**
>
> We thank reviewer La1k for the valuable feedback. Here, we aim to expand the discussion on the comparison between sequence append and channel concatenation. The most critical concern with sequence append approach is its significant increase in sequence length, which is highly undesirable for MLLM due to the following issues:
>
> + Increased training and inference cost: As shown in the table below, longer sequences lead to a substantial rise in both the first forward time and overall training time.
> + Occupying context length and position IDs in the LLM: This may limit the model's ability to handle longer input contexts effectively.
>
> To quantitatively illustrate this, we interpolated the visual tokens to different sequence lengths and measured the first forward time and total training time during the supervised fine-tuning stage. As shown in the table below, increasing the sequence length from 1024 to 3072 results in a 61% increase in first forward time and a 50% increase in training time.
>
> | Model       | Visual Token Number | Total Forward Time | Training Time |
> |-------------|---------------------|--------------------|---------------|
> | Eagle-X5-7B |                1024 |              0.46s | ~24h          |
> | Eagle-X5-7B |                2048 |              0.59s | ~31h          |
> | Eagle-X5-7B |                3072 |              0.74s | ~36h          |
>
> More importantly, as extensively studied in LLM research, increasing the sequence length causes challenges in identifying relevant content—commonly referred to as the "needle in a haystack" problem [1]. As shown in the last two rows of Table 4 in the manuscript, sequence append lags behind channel concatenation in performance.
>
> The increased number of visual tokens also consumes more position IDs, making the generalization of position embeddings difficult [2]. For instance, in our experiments with Vicuna-7B/13B and Yi-34B, the maximum context length is 4096 as defined by the `max_position_embeddings` in their config [3]. This constraint limits the model's ability to accommodate more than three vision encoders with sequence appending, assuming each encoder outputs a sequence length of 1024 in our cases.
>
> 1. BABILong: Testing the Limits of LLMs with Long Context Reasoning-in-a-Haystack, NeurIPS 2024
> 2.  YaRN: Efficient Context Window Extension of Large Language Models, ICLR 2024.
> 3.  https://huggingface.co/lmsys/vicuna-7b-v1.5/blob/main/config.json
>     https://huggingface.co/lmsys/vicuna-13b-v1.5/blob/main/config.json
>    https://huggingface.co/NousResearch/Nous-Hermes-2-Yi-34B/blob/main/config.json

---

> ### Author Response · Authors · 2024-11-21
> **Response to Reviewer La1k Part 1**
>
> Thanks for the constructive feedback! We hope to clarify the confusion and answer the questions below. Feel free to let us know if you have any further questions!  Here we address all the concerns:
> 1) the performance gain of including more vision encoders;
> 2) the throughputs of the vision tower;
> 3) the additional cost of unfreezing the vision tower during training.
>
> __The performance gain from including more vision encoders__
>
> The performance gain after X2, although less significant number-wise, is still very consistent. What’s most important, is the more significant benefit compared to simply scaling a single encoder (for example, InternVL-6B), due to the heterogeneity among expert architectures, the diversity in pre-trained knowledge, and the better coverage of different domain-expertise.
>
> Here’s how we interpret the relatively reduced improvement:
> 1) ConvNeXt itself is indeed a very strong vision encoder. It is convolutional, which well complements the CLIP encoder. It is large (846M in size). And it is pretrained on vision-language alignment tasks. Note that, the relative improvement is influenced by the capability of each encoder. And our round-robin strategy would systematically bias towards earlier rounds of improvement. It is also very natural that everything has a diminishing gain. Considering these factors, the improvements from MoVE is actually very promising.
> 2) The benefit in the expansion of domain-specific expertise and capabilities should not be overlooked. This is particularly true for a generalist model, and the benefits here are sometimes implicit and not fully reflected by the scores on paper. In particular, many additional vision encoders are task-specific. For instance, Pix2Struct helps document and chart understanding, while EVA02 and SAM help object-centric, localization, and open-world recognition tasks. As shown in Table 12 of the appendix, from X3 to X5 where more specialized encoders like Pix2Struct and SAM are added, the improvements are more task-specific—most pronounced on datasets such as VizWiz, DocVQA, and ChartVQA. Although these experts can’t boost the average performance significantly as a strong vision language experts, they can help solving complex real-world tasks with special representations inherited from its own pretraining task. First part of Fig. 5 has listed some of the examples.

---

> ### Author Response · Authors · 2024-11-21
> **Response to Reviewer La1k Part 2**
>
> __Throughput of the vision tower__
>
> For the second concern, we want to first clarify that, for a MLLM, the additionally introduced costs in vision encoder is relatively marginal compared to the LLM auto-regressive generation as a whole. Hence, all of the methods share similar generation time with the same language model and generation length. To provide quantitative evidence, we evaluate image captioning to measure the inference speed of different vision encoders. The experiment uses Vicuna-7B as the base model, employing float16 precision and KV cache optimization for efficient LLM generation. The generation process is stopped after producing 128 tokens (approximately 100 words). The input consists of a 1024 x 1024 resolution image and a prompt—“Describe this image in detail.”—which results in 50 input text tokens plus a chat template.
>
> We calculate four distinct computational metrics:
>
>   - Generation time: The time taken to generate 128 tokens, measured from the input of the model to the output of the final token.
>   - Vision encoder latency: The time required for the vision encoder to process the input image and produce feature maps.
>   - FLOPs of the vision encoder: The total floating-point operations performed by the vision encoder.
>   - FLOPs of the first forward pass: The total operations required to compute the hidden states for both the input image and text embeddings.
>
> These metrics allow us to provide a comprehensive analysis of computational costs and efficiency for the vision encoders within the context of a full MLLM pipeline.
>
> | Model    | Input Resolution | Parameters | Generation Time | Vision Tower Latency | Vision Tower FLOPs | First Forward FLOPs | Avg. |
>  | -------- | ------- | ------- | ------- | ------- | ------- | ------- | ------- |
>  | InternVL  | 448   | 5536M   | 4.01s   | 0.08s   | 5670   | 13037   | 671.5   |
>  | CLIP (X1) High-res  | 448   | 304M   | 3.85s   | 0.02s   | 422   | 7533   | 670.5   |
>  | CLIP + ConvNeXt (X2)  | 1024   | 1155M   | 3.92s   | 0.04s   | 3347   | 10714   | 681.5   |
>  | CLIP + ConvNeXt + EVA-02 (X3)  | 1024   | 1460M   | 3.97s   | 0.07s   | 4281   | 11650   | 690.7  |
>  | Eagle-X4-7B  | 1024   | 1974M   | 4.10s  | 0.17s   | 5307   | 12681   | 694.6   |
>  | Eagle-X5-7B  | 1024   | 2283M   | 4.19s   | 0.24s   | 6617   | 13995   | 697.1   |
>
>   __Avg.__ means the average score across a number of VLM benchmarks.
>
> The results show that the latency of the vision encoder constitutes only 5.7% of the total generation time for the heaviest model, X5. This proportion decreases further with larger language models, such as 3% for a 13B model, and becomes even smaller in scenarios involving longer token generation, such as multi-round conversations or detailed explanations. In terms of floating-point operations (FLOPs), the vision encoder accounts for less than 50% of the operations during the initial forward pass. Meanwhile, the auto-regressive generation process requires repeated computation for each subsequent token, significantly outweighing the vision encoder's contribution to overall computation.
>
> The reviewer mentioned that methods with similar inference cost should be compared for a fair comparison, which we think is very reasonable. We choose the current largest vision encoder, InternVL 6B, as the vision encoder with the same training data and recipe. With similar FLOPs and over 2x parameters, InternVL-6B still falls behind our method.
>
> In addition, we would also like to mention that our current implementation is based on a naive PyTorch setup, where each vision encoder requires stand-alone processing code and is computed sequentially. However, in practice, each vision encoder can be assigned to a separate GPU since they operate in parallel independently. In such a scenario, the inference time for the entire vision encoder ensemble would be largely determined by the slowest vision encoder. For Eagle-X5, this would reduce the total inference time to ~0.10s instead of 0.24s.

---

> > ### Author Response · Authors · 2024-11-21
> > **Response to Reviewer La1k Part 3**
> >
> > __Additional cost of unfreezing the vision tower__
> >
> > Here we list the training time comparison between unfreezing/freezing the vision encoders during supervised fine-tuning. The training is conducted on 32 A100 GPUs across four nodes using DeepSpeed Zero-2 optimization. Each GPU processes a batch size of 4, resulting in a total batch size of 128.
> >
> > As shown in the table below, unfreezing the vision encoder during training increases both training time and memory usage. However, we find this step to be crucial for enhancing performance, particularly when the input resolution changes or when the vision encoder has not been pretrained on vision-language alignment tasks. For example, unfreezing the vision encoder during the second stage improves the average performance of EVA02 (pretrained on detection) from 543.7 to 639.1. Please refer to Table 3 for more detailed comparisons. Considering the performance gain and the fact that unfreezing the vision encoder during training does not increase inference time, we believe the performance gains fully justify the additional training time.
> >
> > | Model | Freeze Vision | Training Time | Memory |
> > | -------- | ------- | ------- | ------- |
> > | Eagle-X5-7B  | True  | ~16h  | 40GB   |
> > | Eagle-X5-7B  | False  | ~24h  | 52GB   |
> > | Eagle-X5-13B  | True  | ~23h  | 56GB   |
> > | Eagle-X5-13B  | False  | ~33h  | 64GB   |
> >
> > __More details about Deformable Attn__
> >
> > We thank Reviewer La1lk for the thoughtful feedback! We will include additional discussion on the motivation and implementation of the deformable attention (DA)-based fusion approach. We incorporated DA as a fusion method for two main reasons:
> >
> > 1) DA is widely adopted in various vision tasks, achieving state-of-the-art performance while maintaining high efficiency.
> > 2) DA shares conceptual similarities with the MiniGemini architecture. It can also be seamlessly integrated.
> >
> > First, DA has become a critical component in many cutting-edge approaches across a range of vision tasks, such as DINO for object detection, Grounding-DINO for visual grounding, and BEVFormer for autonomous driving. Given its proven effectiveness in these applications, we were motivated to explore its potential for enhancing the visual representation within MLLMs.
> >
> > Second, DA closely resembles the patch information mining architecture proposed by the important compared method, Mini-Gemini, where each pixel attends to neighboring pixels to facilitate information fusion. However, DA can dynamically predict the neighborhood, enabling larger and more flexible receptive fields while maintaining high efficiency.
> >
> > Initially, we did not anticipate that simple direct concatenation would already yield strong performance. As a result, we tested several architectures in parallel, with DA considered an enhancement of MiniGemini’s fusion method. Therefore, we included DA in our comparisons to provide a comprehensive evaluation.

---

> ### Author Response · Authors · 2024-11-21
> **Response to Reviewer La1k Part 4**
>
> __Long Sequence__
>
> We thank reviewer La1k for the valuable feedback. Here, we aim to expand the discussion on the comparison between sequence append and channel concatenation. The most critical concern with sequence append approach is its significant increase in sequence length, which is highly undesirable for MLLM due to the following issues:
>
> 1) __Increased training and inference cost__: As shown in the table below, longer sequences lead to a substantial rise in both the first forward time and overall training time.
>
> 2) __Occupying context length and position IDs in the LLM__: This may limit the model's ability to handle longer input contexts effectively.
>
> To quantitatively illustrate this, we interpolated the visual tokens to different sequence lengths and measured the first forward time and total training time during the supervised fine-tuning stage. As shown in the table below, increasing the sequence length from 1024 to 3072 results in a 61% increase in first forward time and a 50% increase in training time.
>
> | Model    | Visual Token Number | Total Forward Time | Training Time |
> | -------- | ------- | ------- | ------- |
> | Eagle-X5-7B  | 1024   | 0.46s   | ~24h |
> | Eagle-X5-7B | 2048    | 0.59s | ~31h |
> | Eagle-X5-7B    | 3072   | 0.74s | ~36h |
>
> More importantly, as extensively studied in LLM research, increasing the sequence length causes challenges in identifying relevant content—commonly referred to as the "needle in a haystack" problem [1]. As shown in the last two rows of Table 4 in the manuscript, sequence append lags behind channel concatenation in performance.
>
> The increased number of visual tokens also consumes more position IDs, making the generalization of position embeddings difficult [2]. For instance, in our experiments with Vicuna-7B/13B and Yi-34B, the maximum context length is 4096 as defined by the "max_position_embeddings" in their config [3]. This constraint limits the model's ability to accommodate more than three vision encoders with sequence appending, assuming each encoder outputs a sequence length of 1024 in our cases.
>
> __Reference__
>
> [1] BABILong: Testing the Limits of LLMs with Long Context Reasoning-in-a-Haystack, NeurIPS 2024
>
> [2] YaRN: Efficient Context Window Extension of Large Language Models, ICLR 2024.
>
> [3] https://huggingface.co/lmsys/vicuna-7b-v1.5/blob/main/config.json
>     https://huggingface.co/lmsys/vicuna-13b-v1.5/blob/main/config.json
>    https://huggingface.co/NousResearch/Nous-Hermes-2-Yi-34B/blob/main/config.json

---

> ### Comment · Reviewer_La1k · 2024-11-21
>
> I thank the authors for their detailed response, and additional results, I am pleased to raise my score to 6.

---

> > ### Author Response · Authors · 2024-11-21
> > **Thank You!**
> >
> > We are glad the response helped to address the concerns. Thank you very much!

---

### Official Review · Reviewer_Y3dK · 2024-11-05

**Soundness:** 3
**Presentation:** 3
**Contribution:** 3
**Rating:** 8
**Confidence:** 4

**Summary:**

This paper conducts a systematic study on the **mixture of vision encoders** in multimodal large language models (LLMs). Specifically, the paper thoroughly compares the performance differences between various combinations of vision encoders and fusion methods, ultimately finding that simply concatenating the tokens from different encoders outperforms more complex fusion strategies. To bridge the gap between different vision encoders, the authors propose a **pre-alignment** strategy, which enhances the subsequent collaboration between vision encoders by aligning them with a small LLM in advance. The proposed method demonstrates significant improvements across several metrics.

**Strengths:**

This paper provides a detailed and systematic study on the **mixture of vision encoders**, offering valuable insights to the research community.

**Weaknesses:**

1. **Limited novelty**:
   While the primary contribution of this paper lies in the extensive comparisons and ablation studies, the method itself lacks significant innovation. Technically, it does not introduce any groundbreaking insights for the community.

2. **Flaws in Experimental Setup**:
   There are some flaws in the experimental setup. In the comparison shown in **Table 5**, the configuration using pre-alignment includes an additional **stage 1** of training compared to the configuration without pre-alignment, leading to an imbalance in the amount of training. It is recommended that the authors add a setup where **stage 1** is removed and **stage 3** is trained for an additional epoch to eliminate the unfair advantage of extra training rounds. Furthermore, since the vision encoders are pre-aligned in **stage 1**, why is it necessary to retrain in **stage 3**? What would happen if the vision encoders were simply frozen?

3. **Computational and Memory Costs**:
   Since the proposed method involves mixing multiple different vision encoders, the computational and memory costs during training and inference are important factors to consider. When comparing with other methods, these indicators should be included. Additionally, it would be helpful to provide the computational cost and memory usage for each combination in **Table 6** for a more comprehensive evaluation of different vision encoder combinations. **Table 6** shows that the performance difference between five encoders and three encoders is minimal. If the cost is significantly higher, is it necessary to use the combination of five encoders?

4. **Omission of a Key Model**:
   **SigCLIP** is one of the widely used vision models for MLLMs. Why is it not included as an option in **Table 2**?

**Questions:**

Please refer to the weaknesses.

---

> ### Author Response · Authors · 2024-11-21
> **Novelty and Contribution of Our Project**
>
> We appreciate the reviewer’s feedback on the contributions of our paper. While we acknowledge that our project does not introduce groundbreaking new architectures, we wish to emphasize that it offers several important conclusions and methods that contribute novel insights to the community that are untouched by prior works. These findings provide a practical blueprint for building high-resolution multimodal LLMs with strong performance across diverse tasks using straightforward combinations of multiple vision encoders, all without requiring large-scale pretraining. Based on the extensive comparisons and ablations, we want to highlight three parts of contributions.
>
> + **Systematic Exploration of Vision Encoder Fusion**
>   + We systematically study the design space of fusing multiple vision encoders, which is missed by prior works. We demonstrate that even simple architectures can effectively combine multiple vision experts. This avoids the need for designing or fine-tuning complex fusion architectures or scaling up vision encoder parameters and pretraining data.
> + **Pre-Alignment to Address Misalignment Issues**
>   + We are the first to identify and address the critical misalignment issue between different vision encoders by proposing a novel pre-alignment strategy.
>
> As acknowledged by Reviewer La1k, our work addresses a critical issue in the development of multimodal LLMs, demonstrating that our approach represents meaningful progress by integrating effective modifications to enhance model coherence and performance.
>
> Additionally, as noted by Reviewer 1gkn, our pre-alignment strategy—where non-text-aligned vision experts are individually fine-tuned with a frozen LLM before joint training—constitutes a valuable contribution to the field. Notably, no prior studies have specifically addressed the pretraining challenges associated with multi-vision encoder setups. Our pre-alignment method effectively addresses this issue, offering a design that is both efficient and effective.
>
> Together, we believe these contributions underscore the impact and utility of our work, providing practical insights and methodological advancements that benefit the broader research community.

---

> ### Author Response · Authors · 2024-11-21
> **Experimental Setup**
>
> Thank the reviewer Y3dK for the questions and the suggestions for more comprehensive experiments! Here we address all the questions and suggestions on the experiment setup.
>
> 1. **Does pre-alignment work by simply extending SFT training with more iterations?**
>
>  we would like to highlight that in Table 7, row 2 of our paper, we present the experimental results of a configuration where stage 1 is removed and stage 3 is instead trained for an additional epoch to balance the total training effort. As shown, even under this modified setup, our model with pre-alignment (see row 5) demonstrates superior performance, underscoring the benefits of incorporating the pre-alignment stage. This result further supports our approach, indicating that the observed performance improvements are not solely due to an imbalance in training rounds and data points but are indeed a result of the pre-alignment's effectiveness.
>
> 2. **Why is it necessary to fine-tune the vision encoders in stage 3?**
>
> In response to your question regarding the necessity of further fine-tuning the vision encoders in stage 3 after their pre-alignment in stage 1, we have conducted an experiment where the vision encoders are frozen in stage 3. The results, shown in the table below, comparing the last two rows of each group, freezing the vision encoders during supervised fine-tuning case significant drop, yet it’s better than freezing the vision encoders without pre-alignment. These results indicate that keeping the vision encoders unfrozen in stage 3 is indeed essential for optimal performance. The plausible reason is that the language models are updated during the SFT, making the vision encoders and language model mis-aligned again. Hence, unfreezing the vision encoders helps to mitigate this misalignment, ensuring better integration and performance.
>
> |     CLIP | Vision Experts  | Unfreeze | Pre-Align | Avg.     |
> |----------|-----------------|----------|-----------|----------|
> | CLIP-448 | SAM-1024        | No       | No        | 630.6    |
> |          |                 | Yes      | No        | 662.5    |
> |          |                 | Yes      | Yes       | 672.3    |
> |          |                 | No       | Yes       | 648.5    |
> | CLIP-448 | ConvNext-1024   | No       | No        | 652      |
> |          |                 | Yes      | No        | 681.5    |
> |          |                 | Yes      | Yes       | 686.2    |
> |          |                 | No       | Yes       | 670.1    |
> | CLIP-448 | Pix2struct-1024 | No       | No        | 653.5    |
> |          |                 | Yes      | No        | 673.7    |
> |          |                 | Yes      | Yes       | 680.4    |
> |          |                 | No       | Yes       | 665.7    |
> | CLIP-448 | EVA-02-1024     | No       | No        | 630.2    |
> |          |                 | Yes      | No        | 659.2    |
> |          |                 | Yes      | Yes       | 668.2    |
> |          |                 | No       | Yes       | 645.2    |

---

> ### Author Response · Authors · 2024-11-21
> **Efficiency Comparison with Other Models**
>
> Thank Reviewer Y3dK for the feedback! Here we discuss the efficiency comparison with other models.
>
> We have provided a detailed model size comparison with other models in the Table below. Our model achieves the best performance across all benchmarks while maintaining a model size smaller than competitors such as InternVL and Monkey. While our model size is slightly larger than some alternatives, it remains within a similar range, balancing performance and efficiency. The full results can be found in Table 6 of our paper.
>
> |      Model  | Size | LLM Size |  MME | OCRBench | TextVQA |
> |:-----------:|:----:|:--------:|:----:|:--------:|:-------:|
> |   llava1.5  | 7.4B |    7B    | 1510 |    297   |   58.2  |
> |  llava-next | 7.4B |    7B    | 1519 |    490   |   64.9  |
> |  MiniGemini | 7.2B |    7B    | 1523 |    477   |   65.2  |
> |    Monkey   | 9.8B |    7B    |   -  |    514   |   67.6  |
> |   Internvl  |  13B |    7B    | 1525 |     -    |   57.0  |
> | Eagle(Ours) | 9.2B |    7B    | **1582** |   **566**   |   **71.9**  |

---

> ### Author Response · Authors · 2024-11-21
> **Efficiency Analysis of Different Vision Encoder Combinations**
>
> We would like to thank Reviewer Y3dK for the valuable feedback. Here, we provide additional analysis on how more vision encoders improve the performance and discuss the associated efficiency trade-offs.
>
> First, regarding the less significant performance gains after incorporating three vision encoders: initially, we add strong vision encoders pre-trained on visual-language alignment, i.e., the ConvNeXt vision encoder, which leads to significant overall performance improvements. Subsequently, we incorporate vision encoders pre-trained on more specialized tasks—for example, Pix2Struct for document and chart understanding, and EVA02 for object perception. However, compared to general-purpose vision encoders, these additional task-specific encoders primarily enhance performance in areas directly related to their pretraining tasks, resulting in less noticeable gains across general benchmarks. This pattern is evident in Table 12 of the appendix. The addition of the general-purpose vision alignment expert, ConvNeXt, yields a substantial performance boost. By contrast, moving from X3 to X5, where specialized encoders like Pix2Struct and SAM are added, the improvements are more task-specific, with the most pronounced gains observed on datasets such as VizWiz, DocVQA, and ChartVQA.
>
> Although these task-specific capabilities is less helpful for the overall performance, they are also important for many real-world tasks. Fig. 5 has given some examples on how these task-specific encoders improve the CLIP + ConvNeXt model. At the same time, we want to emphasize that the primary contribution of our work lies in the systematic investigation process and proposed training strategies rather than the X5 model itself. Users can adapt our methodology to create their own encoder combinations.
>
> We also want to clarify that, incorporating more vision encoders will affect the throughput of the vision encoder alone. However, considering the MLLM as a whole, we only increase the total generation time a little, because the language model and the auto-regressive generation will take up most of the time. The language model will have more parameters and the vision tower only participates in the first forward pass of the auto-regressive iterations. To provide quantitative evidence, we evaluate image captioning to measure the inference speed of different vision encoders. The experiment uses Vicuna-7B as the base model, employing float16 precision and KV cache optimization for efficient LLM generation. The generation process is halted after producing 128 tokens (approximately 100 words). The input consists of an image and a prompt—“Describe this image in detail.”—which results in 50 input text tokens with 1024 visual tokens.
> We calculate four metrics for efficiency analysis:
> + Generation time: The time taken to generate 128 tokens, measured from the input of the model to the output of the final token.
> + Vision encoder latency: The time required for the vision encoder to process the input image and produce feature maps.
> + FLOPs of the vision encoder: The total floating-point operations performed by the vision encoder.
> + FLOPs of the first forward pass: The total operations required to compute the hidden states for both the input image and text embeddings.
>
> |               Model           | Input Resolution | Parameters | Generation Time | Vision Tower Latency | Vision Tower FLOPs |
> |:-----------------------------:|:----------------:|:----------:|:---------------:|:--------------------:|:------------------:|
> |          Eagle-X5-7B          |       1024       |    2283M   |      4.19s      |         0.24s        |        6617        |
> |          Eagle-X4-7B          |       1024       |    1974M   |      4.10s      |         0.17s        |        5307        |
> | CLIP + ConvNeXt + EVA-02 (X3) |       1024       |    1460M   |      3.97s      |         0.07s        |        4281        |
> |      CLIP + ConvNeXt (X2)     |       1024       |    1155M   |      3.92s      |         0.04s        |        3347        |
> |           CLIP (X1)           |        448       |    304M    |      3.85s      |         0.02s        |         422        |
> |            InternVL           |        448       |    5536M   |      4.01s      |         0.08s        |        5670        |
>
> The results show that the latency of the vision encoder constitutes only 5.7% of the total generation time for the heaviest model, X5. This proportion decreases further with larger language models, such as 3% for a 13B model, and becomes even smaller in scenarios involving longer token generation, such as multi-round conversations or detailed explanations. In terms of floating-point operations (FLOPs), the vision encoder accounts for less than 50% of the operations during the initial forward pass. Meanwhile, the auto-regressive generation process requires repeated computation for each subsequent token, significantly outweighing the vision encoder's contribution to overall computation.

---

> ### Author Response · Authors · 2024-12-01
> **Using SigLip as the Vision Encoder**
>
> We thank Reviewer Y3dK for the valuable feedback! In our experiments, we use the CLIP encoder as the baseline LLaVA 1.5 to ensure a fair comparison. Additionally, we note that the compared methods, such as LLaVA-NeXt, LLaVA-UHD, LLaVA-HR, and Mini-Gemini, also utilize the CLIP encoder. However, as you highlighted, SigLip has become a popular choice for MLLMs. In response, we have re-conducted the experiments, substituting the CLIP in our models with the SigLip encoder.
>
> First, using the same experiment setting as Fig. 4 in the main paper, we replaced the CLIP with SigLip and conducted a round-robin vision encoder combination search again. As shown in the table below, when using SigLip as the base model, the model's performance consistently improves as the number of vision experts increases. This finding aligns with our observations when CLIP encoder serves as the base model.
>
> | #Encoder | Encoder   Combination |   GQA   |  MME | OCR | SQA-img |   POPE  | TextVQA | DocVQA  | ChartQA |   SEED  |  VizWiz | RealworldQA |   AI2D  |   Avg.   |
> |:--------:|:---------------------:|:-------:|:----:|:---:|:-------:|:-------:|:-------:|:-------:|:-------:|:-------:|:-------:|:-----------:|:-------:|:--------:|
> |     1    |         SigLip        | 63.9    | 1552 | 544 | 71.6    | 86.2    | 65.2    | 62.7    | 59.6    | 70.5    | 57.0    |   57.9      | 72.2    | 665.6    |
> |     2    |         SL+CN         | 63.6    | 1501 | 535 | 71.3    | 88.2    | 71.4    | 75.7    | 65.6    | 73.0    | 52.1    |   58.8      | 71.8    | 683.3    |
> |     3    |        SL+CN+DI       | 64.9    | 1493 | 507 | 70.5    | 88.0    | 69.5    | 72.7    | 64.0    | 72.5    | 56.3    |   59.4      | 71.8    | 679.0    |
> |     3    |        SL+CN+SA       | 63.3    | 1520 | 532 | 70.3    | 88.1    | 71.5    | 75.2    | 64.3    | 72.4    | 48.6    |   59.2      | 71.6    | 678.1    |
> |     3    |        SL+CN+PS       | 63.6    | 1501 | 535 | 71.3    | 88.2    | 71.4    | 75.7    | 65.6    | 73.0    | 52.1    |   59.4      | 71.8    | 683.8    |
> |     3    |        SL+CN+EV       | 63.5    | 1513 | 536 | 70.6    | 88.4    | 71.2    | 74.7    | 65.6    | 73.0    | 53.5    |   58.6      | 72.0    | 684.4    |
> |     4    |      SL+CN+EV+DI      | 63.8    | 1518 | 534 | 71.1    | 88.8    | 70.9    | 74.2    | 65.9    | 73.0    | 54.3    |   60.3      | 72.1    | 686.3    |
> |     4    |      SL+CN+EV+SA      | 63.4    | 1507 | 524 | 70.2    | 89.2    | 70.6    | 75.6    | 65.2    | 72.6    | 54.8    |   58.3      | 71.6    | 682.6    |
> |     4    |      SL+CN+EV+PS      | 63.7    | 1522 | 537 | 70.4    | 89.3    | 70.2    | 75.0    | 65.8    | 73.3    | 56.7    |   60.6      | 71.8    | 688.8    |
> |     5    |     SL+CN+EV+PS+SA    | 63.8    | 1531 | 540 | 71.1    | 88.8    | 70.6    | 75.6    | 65.4    | 72.9    | 55.7    |   60.0      | 72.3    | 689.0    |
> |     5    |     SL+CN+EV+PS+DI    | 64.0    | 1531 | 533 | 70.8    | 88.4    | 69.9    | 73.9    | 64.3    | 73.1    | 55.0    |   60.4      | 72.5    | 685.1    |
>
> We also substituted the CLIP model in Eagle-X5 with SigLip-L encoder, and the results are presented in the table below. While replacing the CLIP encoder with SigLip encoder leads to some fluctuations across different benchmarks, the overall performance remains very similar. Notably, the pre-alignment strategy yields a more significant performance improvement, further emphasizing the effectiveness and necessity of pre-aligning different vision encoders.
>
> |             Model         |  MME |  MMB  |  SEED | MathVista |  MMMU |  POPE | SQA_i |  GQA  | VizWiz | VQAv2 | OCR | TextVQA | ChartQA |
> |:-------------------------:|:----:|:-----:|:-----:|:---------:|:-----:|:-----:|:-----:|:-----:|:------:|:-----:|:---:|:-------:|:-------:|
> |  X5-7B-Prealign   (CLIP)  | 1582 | 69.7  | 73.7  |   38.2    | 38.0  | 88.7  | 71.9  | 64.6  |  58.7  | 83.6  | 566 |  71.9   |  69.3   |
> | X5-7B-Prealign   (SigLip) | 1547 | 68.2  | 73.6  |   39.3    | 36.0  | 89.4  | 71.2  | 63.9  |  54.8  | 82.9  | 540 |  71.3   |  68.6   |
> |      X5-7B   (SigLip)     | 1531 | 66.8  | 72.9  |   37.5    | 35.7  | 88.8  | 71.1  | 63.8  |  48.2  | 83.3  | 540 |  70.6   |  65.4   |
> |  X5-13B-Prealign   (CLIP) | 1604 | 70.5  | 74.9  |   39.7    | 38.0  | 88.2  | 73.1  | 64.4  |  60.9  | 84.5  | 573 |  73.9   |  71.0   |
> |  X5-13B-Prealign (siglip) | 1575 | 71.2  | 74.7  |   39.5    | 36.2  | 89.1  | 72.7  | 63.9  |  49.9  | 84.2  | 583 |  74.3   |  72.3   |
> |     X5-13B   (SigLip)     | 1527 | 69.8  | 73.8  |   35.7    | 35.0  | 88.6  | 73.6  | 64.6  |  51.2  | 83.8  | 575 |  73.4   |   69.1  |
>
> On the other hand, we also observed that SigLip encoder is not an obvious win over CLIP encoder in our experiments. We believe the main reason is that instead of using 336x336 resolution input, we modify the CLIP encoder to take 448x448 resolution input and unfrozen it during fine-tuning, which leads to a substantial improvment over the original one.

---

> > ### Comment · Reviewer_Y3dK · 2024-12-02
> > **Thanks for your response**
> >
> > Sincerely appreciate the authors for their detailed and comprehensive response. I would like to raise the score since my concerns have been well addressed.

---

> > > ### Author Response · Authors · 2024-12-02
> > > **Thank You!**
> > >
> > > We are glad the response helped to address the concerns. Thank you very much!

---

### Meta-Review · Area_Chair_6eLm · 2024-12-21

**Metareview:**

It is well-known that different vision encoders have different strengths and weaknesses, which has motivated work on VLMs which consider a mixture of different vision encoders. This paper performs a thorough analysis of these mixtures of vision encoders. Reviewers appreciated the thorough experiments and insights, the quality of the writing and the "pre-alignment" proposed by the authors. Reviewers were initially positive about the paper, and the rebuttal phase, where the authors comprehensively addressed the remaining concerns and revised their paper, improved the reviewers impressions even further. Therefore, the decision is to accept the paper.

**Additional Comments On Reviewer Discussion:**

Refer to above. Reviewers were initially positive about the paper, and the rebuttal phase, where the authors comprehensively addressed the remaining concerns and revised their paper, improved the reviewers impressions even further.

---

### Decision · Program_Chairs · 2025-01-22

Accept (Spotlight)